# Rethinking generalization requires revisiting old ideas: statistical mechanics approaches and complex learning behavior

## Abstract

We describe an approach to understand the peculiar and counterintuitive generalization properties of deep neural networks. The approach involves going beyond worst-case theoretical capacity control frameworks that have been popular in machine learning in recent years to revisit old ideas in the statistical mechanics of neural networks. Within this approach, we present a prototypical Very Simple Deep Learning (VSDL) model, whose behavior is controlled by two control parameters, one describing an effective amount of data, or load, on the network (that decreases when noise is added to the input), and one with an effective temperature interpretation (that increases when algorithms are early stopped). Using this model, we describe how a very simple application of ideas from the statistical mechanics theory of generalization provides a strong qualitative description of recently-observed empirical results regarding the inability of deep neural networks not to overfit training data, discontinuous learning and sharp transitions in the generalization properties of learning algorithms, etc.

## 1 Introduction

Neural networks (NNs), both in general (1) as well as in their most recent incarnation as deep neural networks (DNNs) as used in deep learning (2), are of interest not only for their remarkable empirical performance on a variety of machine learning (ML) tasks, but also since they exhibit rather complex properties that have led researchers to quite disparate conclusions about their behavior. For example, some papers lead with the claim that DNNs are robust to a massive amount of noise in the data and/or that noise can even help training (3; 4; 5), while others discuss how they are quite sensitive to even a modest amount of noise (6; 7); some papers express surprise that the popular Probably Approximately Correct (PAC) theory and Vapnik-Chervonenkis (VC) theory do not describe well their properties (7), while others take it as obvious that those theories are not particularly appropriate for understanding NN learning (8; 9; 10; 11; 12; 13); many papers point out how the associated optimization problems are extremely non-convex and lead to problems like local minima, while others point out how non-convexity and local minima are never really an issue (14; 15; 16; 17; 18; 19); some advocate for convergence to flat minimizers (20), while others seem to advocate that convergence to sharp minima can generalize just fine (21); and so on.

These tensions have been known for a long time in the NN area, e.g., see (22; 23; 24; 25; 26; 10; 27; 14), but they have received popular attention recently due to the study of Zhang et al. (7). This recent study considered the tendency of state-of-the-art DNNs to overtrain when presented with noisy data, and its main conclusions are the following.

**Observation 1 (Neural networks can easily overtrain.)** State-of-the-art NNs can easily minimize training error, even when the labels and/or feature vectors are noisy, i.e., they easily fit to noise and noisy data (although, we should note, we found that reproducing this result was not so easy). This implies that state-of-the-art deep learning systems, when presented with realistic noisy data, may always overtrain.

**Observation 2 (Popular ways to regularize may or may not help.)** Regularization (more precisely, many recently-popular ways to implement regularization) fails to prevent this. In particular, methods that implement regularization by, e.g., adding a capacity control function to the objective and approximating the modified objective, performing dropout, adding noise to the input, and so on,

do not substantially improve the situation. Indeed, the only control parameter[1] that has a substantial regularization effect is early stopping.

To understand why this seems peculiar to many people trained in statistical data analysis, consider an SVM, where this does not happen. Let's say one has a relatively-good data set, and one trains an SVM with, say, 90% training accuracy. Then, clearly, the SVM generalization accuracy, on some other test data set, is bounded above by 90%. If one then randomizes, say, 10% of the labels, and one retrains the SVM, then one may overtrain and spuriously get a 90% training accuracy. Textbook discussions, however, state that one can always avoid overtraining by tuning regularization parameters to get better generalization error on the test data set. In this case, one expects the tuned training and generalization accuracies to be bounded above by roughly $90 - 10 = 80\%$. Observation 1 and Observation 2 amount to saying that DNNs behave in a qualitatively different way.

Given the well-known connection between the capacity of models and bounds on generalization ability provided by PAC/VC theory and related methods based on Rademacher complexity, etc. (28; 29), a grand conclusion of Zhang et al. (7) is that understanding the properties of DNN-based learning "requires rethinking generalization." We agree. Moreover, we think this rethinking requires going beyond recently-popular ML methods to revisiting old ideas on generalization and capacity control from the statistical mechanics of NNs (9; 30; 11; 31).

Here, we consider the statistical mechanics (SM) theory of generalization, as applied to NNs and DNNs. We show how a very simple application of it can provide a qualitative explanation of recently-observed empirical properties that are not easily-understandable from within PAC/VC theory of generalization, as it is commonly-used in ML. The SM approach (described in more detail in Sections 2 and A.2) can be formulated in either a "rigorous" or a "non-rigorous" manner. The latter approach, which does not provide worst-case a priori bounds, is more common, but the SM approach can provide precise quantitative agreement with empirically-observed results (as opposed to very coarse bounds) along the entire learning curve, and it is particularly appropriate for models such as DNNs where the complexity of the model grows with the number of data points. In addition, it provides a theory of generalization in which, in appropriate limits, certain phenomenon such as phases, phase transitions, discontinuous learning, and other complex learning behavior arise very naturally, as a function of control parameters of the ML process. Most relevant for our discussion are load-like parameters and temperature-like parameters. While the phenomenon described by the SM approach are not inconsistent with the more well-known PAC/VC approach, the latter is coarse and typically formulated in such a way that these phenomenon are not observed in the theory.

*Our main contribution is to describe a **Very Simple Deep Learning (VSDL) model** that, when viewed from the SM theory of generalization, reproduces Observations 1 and 2.* In this VSDL model, a DNN is a black box, the behavior of which can be controlled by (one or both of) two control parameters, a load-like parameter, denoted $\alpha$, describing the amount of data, perhaps on some scale like the model complexity, and a temperature-like parameter, denoted $\tau$, having to do with noise in the learning process. Importantly, these two parameters can be controlled by a practitioner in very easy ways.

• *Adding noise decreases an effective load.* We model the process of adding noise to the training data as decreasing the magnitude of the effective load-like parameter $\alpha$. This has the effect of decreasing the effective amount of input data relative to the complexity of the NN.

• *Early stopping increases an effective temperature.* We model the iteration complexity of a stochastic iterative training algorithm as an effective temperature-like regularization parameter $\tau$. Performing more iterations and/or decreasing the learning rate corresponds to lower values of $\tau$.

---

[1] By a *control parameter*, we mean a parameter that a practitioner can in practice use to help control the ML process. Well-known examples are the number $n$ of data points, the ratio $n/p$ of the number of data points to the number of features, and the value of $\lambda$ when optimizing objectives of the form $f(x) + \lambda g(x)$. In older NNs, control parameters include the load $\alpha$ on the network, the temperature $\tau$ in the Boltzmann distribution, etc. In modern DNNs, control parameters include parameters characterizing the quality of the data, parameters characterizing the complexity of the model, the ratio of the number of data points to a parameter characterizing the complexity of the model, the amount of dropout, SGD block sizes, learning rate schedules, the number of iterations of an iterative algorithm, etc. Importantly, a control parameter is not just a theoretical parameter which enters into the formalism, e.g., the VC dimension, it is one that can be used operationally to control the output of the learning process.

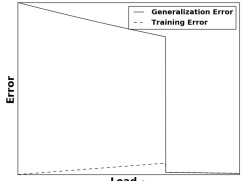 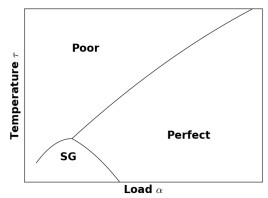 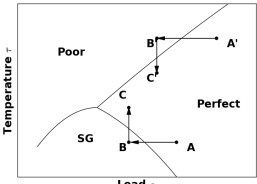

(a) Training/generalization error in the VSDL model.

(b) Learning phases in $\tau$-$\alpha$ plane for VSDL model.

(c) Modeling the process of adding noise to data and adjusting algorithm knobs to compensate.

Figure 1: Schematic of error plots, phase diagrams, and the process of adding noise to input data and then adjusting algorithm knobs for our new VSDL model of classification in DNN learning models. We describe this in Claims 1, 2 and 3 in Section 3.

*We propose that the two parameters used by Zhang et al. (7) (and many others), which are control parameters used to control the learning process, are directly analogous to load-like and temperature-like parameters in the traditional SM approach to generalization.* (Some readers may be familiar with these two parameters from the different but related Hopfield model of associative memory (32; 33), but the existence of two or more such parameters holds more generally (9; 30; 11; 34; 31).)

*Given these two identifications, which are novel to this work, general considerations from the SM theory of generalization, applied even to very simple models like the VSDL model, suggest that complex and non-trivial generalization properties—including the inability not to overfit to noisy data—emerge very naturally, as a function of these two control parameters.* In particular, we note the following (which amount to explaining Observations 1 and 2).

• *One-dimensional phase diagram.* Figure 1(a) illustrates the behavior of the generalization error as a function of increasing (from left to right, or decreasing, from right to left) the load parameter $\alpha$. There is a critical value $\alpha_c$ where the the generalization properties change dramatically, and for other values of $\alpha$ the generalization properties change smoothly.

• *Two-dimensional phase diagram.* Figure 1(b) illustrates the phase diagram in the two-dimensional space defined by the $\alpha$ and $\tau$ parameters. In this figure, the boundaries between different phases mark sharp transitions in the generalization properties of the system, and within a given phase the generalization properties of the system vary smoothly.

• *Adding noise and parameter fiddling.* Figure 1(c) illustrates the process of adding noise to data and adjusting algorithm knobs to compensate. Starting from the $(\alpha, \tau)$ point $A$, which exhibits good generalization behavior, adding noise casues $\alpha$ to decrease, leading to point $B$, which exhibits poor generalization. This can be offset by adjusting (for $A \to B \to C$, this means decreasing) the number of iterations to modify the $\tau$ parameter, again leading to good generalization. Figure 1(c) also illustrates that, starting from the $(\alpha, \tau)$ point $A'$, adding noise casues $\alpha$ to decrease, leading to point $B'$, which also has poor generalization, and this can be offset by adjusting (except for $A' \to B' \to C'$, this means increasing) the number of iterations to modify the $\tau$ parameter to obtain point $C'$.

The VSDL model and these consequences are described in more detail in Sections 3.1 and 3.2.

We should note that the SM approach to generalization can lead to quantitative results, but to achieve this can be technically quite complex (9; 30; 11; 34; 31). Thus, in this paper, we do *not* focus on these technical complexities, lest the simplicity of our main contribution be lost, but we instead leave that for future work. On the other hand, the basic ideas and qualitative results are quite simple, even if somewhat different than the ideas underlying the more popular PAC/VC approach (9; 30; 11; 34; 31).

While it should go without saying, one should of course be careful about naïvely interpreting our results to make extremely broad claims about realistic DNN systems. Realistic DNNs have many more control parameters—the amount of dropout, SGD block sizes, learning rate schedules, the

number of iterations, layer normalization, weight norm constraints, etc.—and these parameters can interact in very complicated ways. Thus, an important more general insight from our approach is that—depending strongly on the details of the model, the specific details of the learning algorithm, the detailed properties of the data and their noise, etc. (which are *not* usually described sufficiently well in publications to reproduce their main results)—going beyond worst-case bounds can lead to a rich and complex array of manners in which generalization can depend on the control parameters of the ML process.

In the next section, Section 2, we will review some relevant background; and then, in Section 3, we will present our main contributions on connecting practical DNN control parameters with load-like parameters, temperature-like parameters, and non-trivial generalization behavior in a VSDL model. In Section A, we will provide a more detailed discussion and explanation of our main result; and in Section 4, we will provide a brief discussion and conclusion.

## 2 BACKGROUND

Here, we will describe some background material that will help to understand our main results.

*A historical perspective.* As historical background, recall that the SM approach to NNs has a long history, indeed, going back to the earliest days of the field (35; 36; 33). For example, following Cowan and Little, in the Hopfield model (32), there is an equivalence between the behavior of NNs with symmetric connections and the equilibrium SM behavior of magnetic systems such as spin glasses, and thus one is able to design NNs for associative memory and other computational tasks (33). Both the SM approach, applied more generally, as well as PAC/VC theory, received a great deal of attention in the 80s/90s (i.e., well before the most recent AI winter) as methodologies to control the generalization properties of NNs. Soon after that time, the ML community turned to methods such as Support Vector Machines (SVMs) and related PAC/VC-based analysis methods that could reduce the ML problem to a related optimization objective in a relatively black-box manner.[2] Subsequent to the renewed interest in DNNs, starting roughly a decade ago (37; 38), most theoretical work within ML has considered this PAC/VC approach to generalization and ignored the SM approach to generalization. There are certain technical[3] and non-technical[4] reasons for this. Here, we will not focus on those reasons, and instead we will describe how the theory of generalization provided by the SM approach to NNs can provide a qualitative description of recently-observed phenomenon. Providing a quantitative description is beyond the scope of this paper, but it is clearly a question of interest raised by our main results.

*PAC/VC versus SM approach to generalization.* For readers more familiar with the PAC/VC approach than the SM approach to learning, most relevant for this paper is that the SM approach highlights that, even for very simple NN systems (not to mention much more complex and state-of-the-art DNN systems), a wide variety of qualitative properties are observable in learning/generalization curves. In particular, as opposed to generalization simply getting gradually and uniformly better with more data, which is intuitive and is suggested[5] by PAC/VC theory, but which empirically is often not the case (40; 27), the SM approach explains (and predicts) why the actual situation can be considerably more complex (9; 30; 10; 11). For example, there can be strong discontinuities in the generalization performance as a function of control parameters; the generalization performance can depend sensitively on details of the model, the specific details of the algorithms that perform approximate computation, the implicit regularization properties associated with these approximate computations, the detailed properties of the data and their noise[6]; etc. This was well-known historically (9; 30; 10; 11); and, as we will describe in more detail below, it is the modern instantiation of these more complex and thus initially-counterintuitive properties that is what researchers have observed in recent years in complex deep learning systems. See Section A.2 for more details.

---

[2]This separation is convenient, basically since it permits one to consider algorithmic optimization questions (i.e., what is the best algorithm to use) separately from statistical inference questions (e.g., as quantified by the VC dimension, Rademacher complexity, or related capacity control measures), but it can be quite limiting.

[3]E.g., it often requires rather strong distribution assumptions and can be technically quite complex to apply.

[4]E.g., it involves taking limits that are quite different than the limits traditionally considered in theoretical computer science and mathematical statistics; and, due to connections with the so-called replica method, it is described as "non-rigorous," at least in its usual method of application. See, however, (10; 11), and also (39).

[5]PAC/VC theory provides smooth upper bounds on quantities related to generalization accuracy, but of course a smooth upper bound on a quantity does not imply that the quantity being upper bounded is smooth.

[6]These "details" are usually *not* described sufficiently well in publications to reproduce their main results.

*Phases, phase transitions, and phase diagrams.* A general aspect of the SM approach to generalization is that, depending on the values chosen for control parameters, we expect (in the appropriate limits) NNs to have different phases and thus non-trivial phase diagrams. Informally, by a phase, we mean a region of some parameter space where the aggregate properties of the system (e.g., memorization/retrieval capabilities in associative memory models, generalization properties in ML models, etc.) change reasonably smoothly in the parameters; by a phase transition, we mean a point of discontinuity in the aggregate properties of the system (e.g., where retrieval or generalization gets dramatically better or worse) under the scaling of the control parameter(s) of the system; and by a phase diagram, we mean a plot in one or more control parameters of regions where the system exhibits qualitatively different behavior.[78] For example, in the Hopfield model of associative memory, depending on the values of the load parameter $\alpha = m/N$ (where $m$ is the number of random memories stored in a Hopfield network of $N$ neurons) and the temperature parameter $\tau$, the system can be in a high-temperature ergodic phase or a spin glass phase (in which the states have a negligible overlap with the memories) or a low-$\tau$ low-$\alpha$ memory phase (which has valleys in configuration space that are close to the memory states); and, as the control parameters $\alpha$ and $\tau$ are changed, the system can change its retrieval properties dramatically and qualitatively. (Unsupervised Holographic associative memories and Restricted Boltzmann machines, as well as supervised models of Multilayer perceptrons, all display unique and non-trivial phase behavior.) Here, we are interested in the generalization properties of NNs, and thus we will be interested in how the generalization properties of NNs change as a function of control parameters of the learning process.

## 3 Qualitatively Explaining Deep Neural Network Behavior

Here, we will present a very idealized (but not *too* idealized to lead to useful insights) model of practical deep learning computations. When viewed through the SM theory of generalization, even this very simple model explains several aspects of the performance of large modern DNNs, including Observations 1 and 2.

### 3.1 Our main model

Our main results are presented in the following three claims. The first claim presents our VSDL model (which is perhaps the simplest model that captures two practical control parameters used in realistic DNN systems); the second claim argues that the thermodynamic limit (where the model complexity diverges with the amount of input data) is an appropriate limit under which to analyze the VSDL model; and the third claim states that in this limit the VSDL model has non-trivial phases of learning (that correspond to Observations 1 and 2).

**Claim 1 (A Very Simple Deep Learning (VSDL) model.)** One can model practical DNN training (including the empirical computations of Zhang et al. (7)) in the following manner. A DNN system implements a function, that can be denoted by $f$, where this function maps, e.g., input images to output labels, i.e.,

$$f : x \to [c],$$

where $x$ denotes the input image and $[c] = \{1, \ldots, c\}$ denotes the class labels, e.g., $\{-1, +1\}$ in the case of binary classification. In the simplest case, the dependence of $f$ on various parameters and hyperparameters of the learning process will not be modeled, with the exception of a load-like parameter $\alpha$ and a temperature-like parameter $\tau$. Thus, the function $f$ implemented by the DNN system depends parametrically (or hyperparametrically) on $\alpha$ and $\tau$, i.e.,

$$f = f(x; \alpha, \tau).$$

***We refer to this model as a Very Simple Deep Learning (VSDL) model.*** Importantly, both the $\alpha$ parameter and the $\tau$ parameter can be easily controlled during the DNN training.

---

[7]Perhaps the most familiar example is provided by water: temperature $T$ and pressure $P$ are control parameters, and depending on the value of $T$ and $P$, the water can be in a solid or liquid or gaseous state. Another example is provided by the Erdős-Rényi random graph model: for values of the connection probability $p < 1/n$, there does not exist a giant component, while for values of the connection probability $p > 1/n$, there does exist a giant component.

[8]In physical applications, the "macroscopic" properties and thus the transitions between regions of control parameter space that have dramatically different macroscopic properties are of primary interest. In statistical learning applications, one often engineers a problem to avoid dramatic sensitivity on parameters, and it is usually the case that the values of the "microscopic" variables (e.g., how to improve prediction quality by 1%) are of primary interest. Since we are interested in rethinking generalization and understanding deep learning, we are primarily interested in macroscopic properties of DNN learning systems, rather than their microscopic improvements.

- **Adding noise decreases an effective load $\alpha$.** *First, we propose that adding noise to the training data—e.g., by randomizing some fraction of the labels (or alternatively by adding noise to some fraction of the data values, by adding extra noisy data to the training data set, etc.)—corresponds to decreasing an effective load-like control parameter.* A justification for this is the following. Assume that we are considering a well-trained DNN model, call it $f$, trained on $m$ data points; and let $N$ denote the effective capacity of the model trained on these data, e.g., by training to zero training loss according to some SGD schedule. Assume also that we then randomize $m_{rand}$ labels, e.g., where $m_{rand} = 0.1m$ if (say) 10% of the labels have been randomized. Then, we can use $m_{eff} = m - m_{rand}$ to denote the *effective* number of data points in the new data set. By analogy with the capacity load parameter in associative memory models, we can define an effective load-like parameter, denoted $\alpha$, to be $\alpha = m_{eff}/N$. In this case, adding noise by randomizing the labels decreases the effective number of training examples $m_{eff}$, but the model capacity $N$ obtained by training is similar or unchanged when this noise is added. Thus, adding noise to the training data has the effect of decreasing the load $\alpha$ on the network. The rationale for this association is that, if we recall that the Rademacher complexity (which can be upper bounded by the growth function, which in turn can be bounded by the VC dimension) is a number in the interval $[0, 1]$ that measures the extent to which a model can (if close to 1) or can not (if close to 0) be fit to random data, then empirical results indicate that for realistic DNNs it is close to 1. Thus, the model capacity $N$ of realistic DNNs scales with $m$, and not $m_{eff}$. Thus, if we then train a new DNN model, call it $f'$, on the new set of $m$ data points, $m_{rand}$ of which have noisy labels, and $m_{eff}$ of which are unchanged, again by training to zero training loss according to some related SGD schedule, then $N' \approx N$. Said another way, if the original problem was satisfiable/realizable, then after randomization of the labels, we essentially have $2^{m_{rand}}$ new binary problems of size $|m - m_{rand}|$, many of which are not "really" satisfiable, in a model of appropriate capacity. Since the model $f'$ has far more capacity than is appropriate for $m - m_{rand}$ labels, however, we overtrain when we compute $f'$.

- **Early stopping increases an effective temperature $\tau$.** *Second, we observe that the iteration complexity within a stochastic iterative training algorithm is an effective temperature-like control parameter, and early stopping corresponds to increasing this effective temperature-like control parameter.* A justification for this is the following. Since DNN training is performed with SGD-based algorithms, from a SM perspective, there is a natural interpretation in terms of a stochastic learning algorithm in which the weights evolve according to a relaxational Langevin equation. (Alternatively, see, e.g., (41; 42; 43).) Thus, from the fluctuation-dissipation theorem, there is a temperature $\tau$ that corresponds to the learning rate of the stochastic dynamics.[9] Operationally, this $\tau$ has to do with the annealing rate schedule of the SGD algorithm, which decreases the variability of, e.g., the NN weights, and thus with $t_*^{-1}$, where $t_*$ is the number of steps taken by the stochastic iterative algorithm when terminated. This temperature-like parameter will be denoted by $\tau$; and it depends on $t_*$ as $\tau = \tau(t_*^{-1})$ in some manner that we won't make explicit.

This VSDL model ignores other "knobs" that clearly have an effect on the learning process, but let's assume that they are fixed. Fixing them simply amounts to choosing a potentially sub-optimal value for the other knobs. Doing so does not affect our main argument, it provides a more parsimonious description than needing many knobs, and our main argument can be extended to deal with other control knobs. The point is that both $\alpha$ and $\tau$ are parameters that the practitioner can use to control the learning process, e.g., by adding noise to the input data or by early-stopping.

Of course, $f$ also has a VC dimension, a growth function, an annealed entropy, etc., associated with it. We are not interested in these quantities since they are not parameters that can practically be used to control the learning process, and since the bounds provided by them provide (at best) very little insight into the NN/DNN learning process (e.g., they are so large to be of no practical value, and they often don't even exhibit the correct qualitative behavior) (10).

**Claim 2 (Appropriate limits to consider in the analysis.)** When performing computations on modern DNNs, e.g., as with those of Zhang et al. (7), one trains in such a way that effectively lets the model complexity grow with the number of parameters. *Thus, when considering the VSDL model, one should consider a thermodynamic limit, where the hypothesis space $\mathcal{F}_N$ and the number of data points $m$ both diverge in some manner (as opposed to the limit where one fixes the hypoth-*

---

[9]If one manually fixes $\tau$, e.g., by setting $\tau = 1$ in every equation, there is an *effective temperature* defined, e.g., by the variability of the weight vector norm, and similar considerations hold. Relatedly, certain weight regularization schemes, e.g., weight norm regularization, also serve to provide a form of temperature control.

*esis space $\mathcal{F}$ and lets $m$ diverge), as with the SM approach to generalization, and in contrast to the PAC/VC approach to generalization.* Many of the technical complexities associated with the SM approach to generalization, which are described in detail in the references cited in Section A, are associated with subtleties associated with this limit.

**Claim 3 (Phases of learning and transitions between different phases.)** *Given these identifications, general considerations from the SM theory of generalization—e.g., (9; 30; 10; 11; 31)—directly imply the following hold for models such as the VSDL model in the thermodynamic limit.*

• **One-dimensional phase diagram.** As a function of the load-like parameter $\alpha$, e.g., for $\tau = 0$, one should expect the VSDL model to have error plots that look qualitatively like the ones shown in Figure 1(a). In this figure, the generalization and training errors are plotted as a function of $\alpha$.

• **Two-dimensional phase diagram.** In addition, when the temperature-like parameter $\tau$ is also varied, one should expect the VSDL model to have a phase diagram that looks qualitatively like the one shown in Figure 1(b). In this figure, the $\tau$-$\alpha$ plane is shown, and rather than plotting a third axis to show the generalization error and training error as a function of $\tau$ and $\alpha$, we instead show lines between different phases of learning.

To understand these diagrams, consider first Figure 1(a). Observe that, for $\tau = 0$ as in that figure, as one increases $\alpha$ from a small value (i.e., as one obtains more data), the generalization error decreases gradually (as is intuitive), and then as one passes through a critical value $\alpha_c$ it decreases rather dramatically. Alternatively, as one decreases $\alpha$ from a large value (e.g., as one adds noise to the data), the generalization error increases gradually (again, as is intuitive) and then as one passes through the critical value $\alpha_c$ it increases rather dramatically. The transition from $\alpha > \alpha_c$ to $\alpha < \alpha_c$ means that there is a dramatic increase in generalization error, where one can fit the training data well, but where one does a very poor job fitting the test data. This is illustrated pictorially along the $\tau = 0$ axis of Figure 1(b). These observations hold for any given value of $\tau$, e.g., $\tau = 0$ or $\tau > 0$, although the value $\alpha_c$ may depend on $\tau$. This is shown more generally in Figure 1(b). Moreover, for certain values of $\tau$ greater than a critical value, i.e., for $\tau > \tau_c$, the sharp transition in learning as a function of $\alpha$ may disappear, in which case the system exhibits only one phase of learning.

*Given these figures, the process of adding noise to data and adjusting algorithm knobs to compensate has the following natural interpretation.*

• **Adding noise and parameter fiddling.** See Figure 1(c) for a pictorial representation, in the $(\alpha, \tau)$ plane, of the the process of adding noise to data and adjusting algorithm knobs to compensate. Assume that a DNN is trained to a point $A$, with parameter values $(\alpha_A, \tau_A)$, and assume that at this point the system exhibits good generalization behavior, e.g., as for $\alpha > \alpha_c$ in Figure 1(a). If then some fraction of the data labels are randomly changed, the system moves to point $B$, with parameter values $(\alpha_B, \tau_B)$, where $\tau_B = \tau_A$. At point $B$, the DNN can still be trained to fit the new noisy data, but if enough of the data have their labels changed, then the effective load parameter $\alpha < \alpha_c$, for this value of $\tau$. In this case, the generalization properties on new noisy data are much worse, as for $\alpha < \alpha_c$ in Figure 1(a). Of course, this could then be compensated for by adjusting the temperature parameter $\tau$, e.g., by performing early stopping[10], after which the system moves to $C$, with parameter values $(\alpha_C, \tau_C)$, where $\alpha_C = \alpha_B$. For this new point, if the iterative algorithm is stopped properly, then $\alpha > \alpha_c$ for this new value of $\tau$, in which case the generalization properties are then much better, as for $\alpha > \alpha_c$ in Figure 1(a).

## 3.2 Consequences of our main model

There are many consequences of our VSDL model for NN/DNN learning. Many are technically complex or of quantitative interest, and so we leave them for future work. Here we focus just on their consequences for Observations 1 and 2, both of which follow from Claim 3.

**Conclusion 1 (Neural networks can easily overtrain.)** For realistic NNs and DNNs as well as the VSDL model, there typically is *not* a global control parameter like the Tikhonov value off $\lambda$ or the number $k$ of vectors to keep in the TSVD that permits control on generalization for any phase. This is in contrast to linear or linearizable learning, and it is discussed in more detail in Section A.5. That is, *for certain values of $\tau$ and $\alpha$, which are the parameters used to control the learning process, the system is in a phase where it can't not overfit.* (This is simply Observation 1.)

---

[10]Alternatively, many of the other control parameters used in realistic DNNs have a similar effect.

**Conclusion 2 ((Popular ways to implement) regularization may or may not help.)** Of course, for realistic NNs and DNNs as well as the VSDL model, the number of iterations $t_*$ is a control parameter that can prevent this overfitting, i.e., it is a regularization parameter. That is, for a given value of $\alpha$, i.e., for a given value of the noise added to the data, the only control parameter that can prevent overfitting is $\tau$. That is, *in this idealized model of realistic DNNs, where $\tau$ and $\alpha$ are the two control parameters, for a given amount of effective data, the only way to prevent overfitting is to decrease the number of iterations.* (This is simply Observation 2.)

That is, given our three main claims, our two motivating observations follow immediately.

In a sense, these conclusions here complete our stated objective: revisiting old ideas in the SM of NNs provides a powerful way to rethink the qualitative properties of generalization and to understand the properties of modern DNNs.[11] While this approach—both the VSDL model and the SM theory of generalization—is quite different than the PAC/VC approach that has been more popular in ML in recent years, and while it can be technically complex to apply this approach, our observations suggest that there is value in revisiting these old ideas in greater detail. We could, at this point, simply suggest that the reader read the fundamental material, e.g., (9; 30; 11; 31) and references therein, for more details. Since this literature can be somewhat impenetrable, however, we will in Section A provide a few highlights that are most relevant for understanding our main results.

## 4  DISCUSSION AND CONCLUSION

The approach we have adopted to rethinking generalization is to ask what is the simplest possible model that reproduces non-trivial properties of realistic DNNs. In the VSDL model, we have idealized very complex DNNs as being controlled by two control parameters, one describing an effective amount of data or load on the network (that decreases when noise is added to the input), and one with an effective temperature interpretation (that increases when algorithms are early stopped). Using this model, we have explained how a very simple application of ideas from the SM theory of generalization provides a strong qualitative description of recently-observed empirical results regarding the inability of DNNs not to overfit training data, discontinuous learning and sharp transitions in the generalization properties of learning algorithms, etc.

As we were writing up this paper, we became aware of recent work with a similar flavor (44; 45; 46). In (45), the authors consider a more refined scale-sensitive analysis involving a Lipshitz constant of the network, and they make connections with margin-based boosting methods to scale the Lipshitz constant. In (46), the authors use Information Bottleneck ideas to analyze how information is compressed early versus late in the running of stochastic optimization algorithms, when training error improves versus when it does not. These lines of work provide a nice complement to our approach, and the connections with our results merit further examination.

To conclude, it is worth remembering that these types of questions have a long history, albeit in smaller and less data-intensive situations, and that revisiting old ideas can be fruitful. Indeed, recent empirical evidence suggests the obvious conjecture that "every" DNN has, as a function of its control parameters, some kind of *generalization phase diagram*, as in Figures 1(a) and 1(b); and that fiddling with algorithm knobs has the effect of moving around some kind of parameter space, as in Figure 1(c). In these diagrams, there will be a phase where generalization changes gradually, roughly as PAC/VC-based intuition would suggest, and there will also be a "low temperature" spin glass like phase, where learning and generalization break down, potentially dramatically. At this point, it is hard to evaluate this conjecture, not only since existing methods tend to conflate (algorithmic) optimization and (statistical) regularization issues (suggesting we should better delineate the two in our theory), but also since empirical results are very sensitive to the many knobs and are typically non-reproducible.

---

[11]By the way, in addition to providing an "explanation" of the main observations of Zhang et al. (7), the VSDL model and the SM approach provides an "explanation" for many other phenomena that are observed empirically: e.g., strong discontinuities in the generalization performance as a function of control parameters; that the generalization performance can depend sensitively on details of the model, details of the algorithms that perform approximate computation, the implicit regularization properties associated with these approximate computations, the detailed properties of the data and their noise, that the generalization can decay in the asymptotic regime as a power law with an exponent other than 1 or $1/2$, or with some other functional form, etc.

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

## A    EXPLANATION OF THE QUALITATIVE EXPLANATION

Since to read the fundamental material (9; 30; 11; 31) can be challenging for the newcomer, in this section, we will go into more detail on several very simple models that capture certain aspects of realistic large DNNs. These models have been studied in detail with the SM approach, and they can be used to understand why we said that "general considerations from the SM theory of generalization" imply that the generalization behavior of the VSDL model will exhibit the behavior illustrated in Figure 1. In Section A.1, we describe several very simple models of multilayer networks, all of which have properties consistent with Observations 1 and/or 2 and the discontinuous generalization properties shown in Figures 1(a) and 1(b); in Section A.2, we provide an overview of the PAC/VC versus SM approach to generalization; in Section A.3, we explain the root of these discontinuous generalization properties in an even simpler model which can be analyzed in more detail; in Section A.4, we describe evidence for this in larger more realistic DNNs; and in Section A.5, we review popular mechanisms to implement regularization and explain why they should *not* be expected to be applicable in situations such as we are discussing.

### A.1    VERY SIMPLE MODELS OF MULTILAYER NETWORKS

What, one might ask, are the "general considerations from the SM theory of generalization" that suggest that one should expect a generalization diagram that looks qualitatively like the one shown in Figures 1(a) and 1(b)?

To begin to answer this question, we start by considering three very simple network architectures: the fully-connected committee machine, the tree-based parity machine, and the one-layer reversed-wedge Ising perceptron. We start with these since these are perhaps the simplest examples of networks that capture multilayer and non-trivial representation capabilities, two properties that are central to the success of modern DNNs. It is known that multilayer networks are substantially stronger in terms of representational power than single layer networks, and the fully-connected committee machine and tree-based parity machine represent in some sense two extreme cases of connectivity (47; 34; 48). Also, while the one-layer reversed-wedge Ising perceptron consists of only a single layer, it has a non-trivial activation function that may be viewed as a prototype model for the representation ability of more realistic networks (49; 50).

• *Fully-connected committee machine.* This model is a multi-layer network with one hidden layer containing $K$ elements, and thus it is specified by $K$ vectors $\{J_k\}_{k=1}^K$ connecting the $N$ inputs $S_i$, for $i = 1, \ldots, N$, with the hidden units $H_i$, for $i = 1, \ldots, K$. Then, given any input vector $S \in \mathbb{R}^N$, the activity of the $k^{th}$ hidden unit is given by

$$H_k = \text{sign}\left(\frac{1}{\sqrt{N}} J_k S\right), \quad \text{for } k = 1, \ldots, K,$$

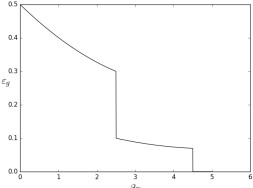 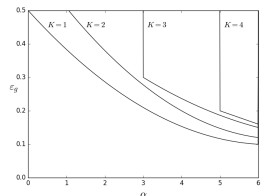 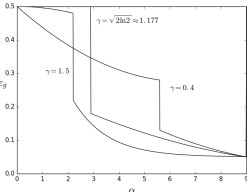

(a) Learning curve for fully-connected committee machine.
(b) Learning curve for tree-based parity machine.
(c) Learning curve for one-layer reversed-wedge Ising perceptron.

Figure 2: Learning curves for classification problems, using three types of network architectures, all exhibiting qualitatively similar continuous-then-discontinuous generalization behavior.

and given the hidden unit vector $H \in \mathbb{R}^K$, the output is given by

$$\sigma = \text{sign}\left(\frac{1}{\sqrt{K}} \sum_{k=1}^{K} H_k\right),$$

i.e., by the majority vote of the hidden layer. See (51; 52; 53) for more details on this model; and see Figure 2(a) for the corresponding learning curve. Figure 2(a) shows the generalization error $\varepsilon$ as a function of the control parameter $\alpha\beta$,[12] where $\alpha$ is a load parameter and $\beta$ is a temperature parameter, and it illustrates the discontinuous behavior of $\varepsilon$ as a function of $\alpha\beta$.

● *Tree-based parity machine.* This model is also a multi-layer network that is also represented by $K$ vectors $\{J_k\}_{k=1}^K$ connecting the $N$ inputs $S_i$, for $i = 1, \ldots, N$, with the hidden units $H_i$, for $i = 1, \ldots, K$, except that here the hidden units have a tree-like structure. Then, given the hidden unit vector $H \in \mathbb{R}^K$, the output is given by

$$\sigma = \prod_{k=1}^{K} H_k,$$

i.e., by the parity of the hidden units. See (54; 55; 47) for more details on this model; and see Figure 2(b) for the corresponding learning curve. Figure 2(b) shows the generalization error $\varepsilon$ as a function of the control parameter $\alpha$, for several different values of $K$, and it illustrates the discontinuous behavior of $\varepsilon$ as a function of $\alpha$.

● *One-layer reversed-wedge Ising perceptron.* This model is a single layer network, but it is interesting since it has a non-trivial activation function. Given any input vector $S \in \mathbb{R}^N$, and the set of weights given by a vector $J \in \mathbb{R}^N$, if we define $\lambda = \frac{1}{\sqrt{N}} \sum_{i=1}^{N} J_i S_i$, then the output classification rule is given by

$$\begin{aligned}
\sigma &= \text{sign}\left((\lambda - \gamma)\lambda(\lambda + \gamma)\right) \\
&= \begin{cases} 1 & \text{if } \lambda \in [-\gamma, 0) \cup [\gamma, \infty) \\ -1 & \text{otherwise} \end{cases},
\end{aligned}$$

where $\gamma$ is a parameter. The non-monotonicity of the activation function is a simple model of representation ability, and the classification is $+1$ or $-1$ depending on the value of $\lambda$ with respect to the value of $\gamma$. See (49; 50) for more details on this model; and see Figure 2(c) for the corresponding learning curve. Figure 2(c) shows the generalization error $\varepsilon$ as a function of the control parameter $\alpha$, for several different values of $\gamma$, and it illustrates the discontinuous behavior of $\varepsilon$ as a function of $\alpha$.

*The point is that in all three of these cases (Figs. 2(a), 2(b), and 2(c)), there is an abrupt change in the learning curve as a function of a load-like parameter.* In addition, while there may be another parameter, e.g., the number $K$ of intermediate groups in the tree-based parity machine or the value

---

[12]In this model, $\alpha\beta$, rather than just $\alpha$, is often used as a control parameter; see (51; 52; 53) for a discussion.

of $\gamma$ in the one-layer reversed-wedge Ising perceptron, there is a range of values of this parameter for which the basic discontinuous generalization behavior is still observed (although there may also be values of that parameter for which the discontinuous generalization behavior is destroyed). *Finally, far from being a peculiarity or a pathology, such behavior is ubiquitous; see, e.g., (56; 41; 57; 58; 59; 60; 61; 62; 63; 64; 65; 66; 67; 68; 69; 70). That is, nearly any non-trivial model exhibits this behavior in the appropriate (thermodynamic) limit.*

To explain the mechanism for this behavior, we will discuss in detail (in Section A.3) two even simpler models, each of which can be analyzed from two complementary approaches to the SM theory of generalization. To do this, we will first review (in Section A.2) two different approaches to understanding generalization in ML.

## A.2 PAC/VC versus SM approach to generalization

For simplicity, let's consider the classification of the elements of some input space $X$ into one of two classes, $\{0, 1\}$. There is a *target rule*, $T$, which is one particular mapping of the input space into the set of classes, as well as a *hypothesis space*, $\mathcal{F}$, which consists of the available mappings $f$ used to approximate the target $T$. The set $\mathcal{F}$ could, e.g., consist of NNs with a given structure. Given this setup, the problem of learning from examples is the following: on the basis of the classification determined by the target rule $T$ for the elements of a subset $\mathcal{X} \subset X$, which is the *training set*, select an element of $\mathcal{F}$ and evaluate how well that element approximates $T$ on the complete input space $X$. One can think of the target rule $T$ as being a *teacher*, and the goal of the *student* is to approximate the teacher as well as possible. The *generalization error* $\varepsilon$ is the probability of disagreement between the student/hypothesis and teacher/target on a randomly chosen subset of $X$.

In the usual setup, one iterates the following process: the student starts with an initial mapping $f^0$ and iterates the following process to obtain a mapping $f^*$: the student is presented an element $x_t \in X$, as well as the teacher's label, $T(x_t)$; and, given that pair $(x_t, T(x_t))$, as well as $f^{t-1}$, the student must construct a new mapping $f^t$ according to some learning rule. If $T \in \mathcal{F}$, then the problem is called *realizable*; otherwise, the problem is called *unrealizable*. For simplicity, let's consider the realizable case (but, of course, this can be generalized). In this case, at a given time step $t$ in the iterative learning algorithm, the *version space* is the subset of $X$ that is compatible with the data/labels thus far presented, i.e.,

$$V(\mathcal{S}) = \{h \in \mathcal{F} : \forall (x, T(x)) \in \mathcal{S}, h(x) = T(x)\},$$

where $\mathcal{S} = (\mathcal{X}, T(\mathcal{X}))$ represents the data seen so far. As an idealization, the zero-temperature Gibbs learning rule is sometimes considered: consider the generalization error of a vector $f$ drawn at random from $V(S)$. The quality of the performance of the student on the training set can be quantified by the *training error* $\varepsilon_t$, which is the fraction of disagreements between the student and teacher output on inputs in the training set (where in the realizable situation one may achieve $\varepsilon_t = 0$). Then, one is interested in characterizing the difference between training error and the generalization error

$$|\varepsilon_t - \varepsilon|. \tag{1}$$

The behavior of Eqn. (1) as a function of control parameters of the learning process is known as the *learning curve*. There are two basic ways to proceed to understand the properties of Eqn. (1).

• *PAC/VC approach to generalization.* One could view the training set size $m$ as the main control parameter, and one could fix the function class $\mathcal{F}$ and other aspects of the setup and then ask how Eqn. (1) varies as $m$ increases. In this case, it is natural to consider

$$\gamma = \mathbf{Pr}\left[|\varepsilon_t - \varepsilon| > \delta\right] \tag{2}$$

as a function of control parameters of the learning process. This is the *probably approximately correct* (PAC) framework (71), where two accuracy parameters, $\delta$ and $\gamma$ are used. In this case, the problem of deciding, on the basis of a small training set, which hypothesis will perform well on the complete input is closely related to the statistical problem of convergence of frequencies to probabilities (28). If one were interested in the $m \to \infty$ limit, one could consider a law of large numbers or a central limit theorem; and if one were interested in learning for finite values of $m$, one might hope to use a Hoeffding-type approach. This approach would provide bounds of the form

$$\mathbf{Pr}\left[|\varepsilon_t - \varepsilon| > \delta\right] \leq 2e^{-2m\delta^2}, \tag{3}$$

but it is *not* appropriate since the rule $f^*$ is not independent of the training data (since the latter is used to construct the former). One way around this is to fix $\mathcal{F}$ and construct a *uniform bound* over the entire hypothesis space $\mathcal{F}$ by focusing on the worst-case situation:

$$\mathbf{Pr}\left[\max_{h \in \mathcal{F}} |\varepsilon_t(h) - \varepsilon(h)| > \delta \right] \leq 2|\mathcal{F}|e^{-2m\delta^2}. \tag{4}$$

It is straightforward to derive this from the Hoeffding inequality if $|\mathcal{F}|$ is finite (where $|\mathcal{F}|$ denotes the cardinality of the set $\mathcal{F}$); and Sauer and Vapnik and Chervonenkis showed that similar results could be obtained even if $|\mathcal{F}|$ is infinite, if the classification diversity of $\mathcal{F}$ is not too large. The most well-studied variant of this uses the so-called *growth function* and the related *VC dimension* $d_{VC}$ of $\mathcal{F}$. Within this PAC/VC approach, minimizing the empirical error within a function class $\mathcal{F}$ on a random sample of $m$ examples leads to a generalization error that is bounded above by $\tilde{O}(d_{VC}/m)$ or $\tilde{O}(\sqrt{d_{VC}/m})$, if the problem is realizable or unrealizable, respectively (28). Note that this power law decay, depending on a simple inverse power of $m$, arises due to the demand for uniform convergence within this approach. Importantly, the only problem-specific quantity in these bounds is the VC dimension $d_{VC}$, which measures the complexity of $\mathcal{F}$, i.e., the learning algorithm, the target rule, etc., do not appear; and these bounds are "universal," in the sense (described in more detail below) that they hold for any $\mathcal{F}$, for any input distribution, and for any target distribution.[13][14]

• *SM approach to generalization.* Alternatively, one could imagine that the function class $\mathcal{F} = \mathcal{F}_N$ varies with the training set size $m$, e.g., as it does in practical DNN learning, and then within the theory let both $m$ and (the cardinality of) $\mathcal{F}_N$ diverge in some well-defined manner. This particular limit is sometimes referred to as the *thermodynamic limit*, and it is common in information theory, error correcting codes, etc. (75). Importantly, this thermodynamic limit is not some arbitrary limit, but instead it is one—when it exists—in which certain quantities related to the generalization error can be computed relatively-easily. Thus, it provides the basis for the SM approach to generalization. (The SM approach to generalization, as opposed to the use of SM in, e.g., associative memory models (32; 33), was first proposed in (76; 25). For accessible introductions, see (11) (for a mathematical statistics or "rigorous" perspective) and (34) (for a statistical physics or "non-rigorous" perspective). See (9; 30; 31) for more comprehensive introductions; and see also (10) for an interesting discussion of the use of SM for cross validation.) One should think of this approach as attempting to describe the learning curve of a parametric class of functions. For example, let's say we want to perform the classification of the elements of some input space $X$ into one of two classes, $\{-1, +1\}$. Then, if we let $\mathcal{F}_1, \mathcal{F}_2, \ldots, \mathcal{F}_N, \ldots$ be a sequence of classes of functions, e.g., NNs trained in some manner with larger and larger data sets, and if for each class $\mathcal{F}_N$ we choose a fixed target function $f_N \in \mathcal{F}_N$, then this leads to a sequence of target functions $f_1, f_2, \ldots, f_N, \ldots$. Of course, it may be that no limiting behavior exists (e.g., in so-called mean field spin glass phases), and in this case the SM approach provides trivial or vacuous results, or more sophisticated variants must be considered. On the other hand, if the limit does exist, then the number of functions in the class at a given error value may have an asymptotic behavior in this limit. In that case, this limit can be exploited by describing the learning curves as a "competition" between the error value (an *energy* term) and the logarithm of the number of functions at that given error value (an *entropy term*). Clearly, if we fix the sample size $m$ and let $N \to \infty$ (respectively, if we fix $N$ and let $m \to \infty$), the we should not expect a non-trivial result, since the function class sizes (respectively, the sample size) are becoming larger but the sample size (respectively, function class sizes) is fixed. Thus, one typically considers the case that $m, N \to \infty$ such that $\alpha = m/N$ is a fixed constant.[15] This $\alpha$ is analogous to the *load on the network* in associative memory models, and it is a control parameter. It lets one investigate the generalization error when the sample size is, e.g., half or twice the number of parameters, which is the approach often adopted in practice. There are two complementary approaches to the SM theory of generalization—see (9; 30; 31) and (10; 11), respectively—and these will be described in Section A.3 for a simple problem.

---

[13]More sophisticated variants of these results exist (using annealed or VC entropy methods (72; 73), Rademacher complexity methods (74), data-dependent VC methods (12), etc.), but they provide bounds of the same form.

[14]Note that, in PAC/VC theory, the number of training points that can be classified exactly is VC dimension, and thus the Zhang et al. (7) results suggest that the VC dimension is (effectively) unbounded.

[15]In the case of least-squares, $\min_x \|Ax - b\|$, for an $n \times p$ matrix $A$, this amounts to considering the limit $n, p \to \infty$, for $\alpha = n/p$ a fixed constant, as opposed to $n$ getting large for fixed $p$, or $p$ getting large for fixed $n$.

### A.3  EXPLANATION IN EVEN MORE SIMPLE NETWORKS: PAC/VC VERSUS SM APPROACHES FOR TWO VERY SIMPLE MODELS

What, one might ask, is the mechanism for the behavior observed in Figs. 2(a), 2(b), and 2(c) for the committee machine, parity machine, and reversed-wedge Ising perceptron (and for many other problems in the SM approach to generalization (9; 30; 11; 31))?

To answer this question, we will consider in some detail two even simpler models: that of a continuous versus a discrete variant of a simple one-layer perceptron. While extremely simple, these two models illustrate the key issue. We emphasize that the behavior to be described has been characterized in three complementary ways: from the rigorous analysis of (11), from extensive numerical simulations, and from the non-rigorous replica-based calculations from statistical physics (56; 77; 9; 59). See Sections IV, V.B and V.D of (9), Section 2 of (11), and Chapters 2 and 7 of (31) for the closest description of what follows.

**The two basic models.**  Given any input vector $S \in \mathbb{R}^N$, the basic single-layer perceptron has a set of weights given by a vector $J \in \mathbb{R}^N$, and the output classification rule is given by

$$\sigma = \text{sign}\,(J \cdot S) = \text{sign}\left(\sum_{i=1}^{N} J_i S_i\right).$$

That is, the classification is $+1$ or $-1$ depending on whether the angle between $S$ and $J$ is smaller or larger than $\pi/2$. In this simple case, the lengths of $S$ and $J$ do not effect the classification, and thus it is common to choose the normalization as $S^2 = \sum_{i=1}^{N} S_i^2 = N$ and $J^2 = \sum_{i=1}^{N} J_i^2 = N$. (In particular, this means that both vectors lie on the surface of an $N$-dimensional sphere with radius $\sqrt{N}$, the surface area of which, $\Omega_0 = \exp\left(\frac{N}{2}\left[1 + \ln(2\pi)\right]\right)$, is, to leading order, exponential in $N$.) In addition, if the inputs are chosen randomly, then the probability of disagreement between $T$ and $J$, which is precisely the generalization error $\varepsilon$, is given by $\varepsilon = \theta/\pi$, where $\theta$ is the angle between $T$ and $J$. If we define the overlap parameter

$$R = \frac{1}{N} J \cdot T = \frac{1}{N} \sum_{i=1}^{N} T_i J_i = \cos(\pi\varepsilon),$$

then the generalization error can be written as

$$\varepsilon = \frac{1}{\pi} \arccos(R).$$

That is, the generalization error depends only on the overlap $R$ between $J$ and $T$. (To set the scale of the error: $\varepsilon = 0$ for $R = +1$; $\varepsilon = 0.5$ for $R = 0$; and $\varepsilon = 1$ for $R = -1$.)

Here are the two basic versions of the perceptron we will consider.

• *Continuous perceptron.* In this model, $J \in \mathbb{R}^N$, subject to the constraint that $J^2 = N$ (and the output $\sigma \in \{-1, +1\}$). In particular, the weights $\{J_i\}_{i=1}^{N}$ are continuous, and they lie on the $N$-dimensional sphere with radius $\sqrt{N}$. This version corresponds to the original perceptron model studied by Rosenblatt (78) (and it is well-described by PAC/VC theory).

• *Ising perceptron.* In this model, $J \in \mathbb{R}^N$, subject on the constraint that each $J_i \in \pm 1$ (and the output $\sigma \in \{-1, +1\}$). That $J \in \{-1, +1\}^N$ implies that $J^2 = N$, but this is a much stronger condition, since they lie on the corners of an $N$-dimensional hypercube. This stronger discreteness condition has subtle but very important consequences. This version was first studied by (79; 56; 77) (and it exhibits the phase transition common to all spin glass models of NNs, and it is not well-described by PAC/VC theory).

In this case, the generalization error $\varepsilon$ decreases as the training set size increases since more and more vectors $J$ become incompatible[16] with the data $\{(x_i, T(x_i))\}_{i=1}^{m}$. To quantify the probability that a vector $J$ remains compatible with the teacher when a new example is presented, we can group the vectors $J$ into classes depending on their overlap $R$ with $T$, i.e., depending on the average

---

[16]For simplicity, we restrict ourselves to realizable problems; but similar results hold in general (9; 30; 11; 31).

generalization error $\varepsilon$. For all $J$ with overlap $R$ (or generalization error $\varepsilon$), the chance of producing the same output as $T$ on a randomly chosen input is (by definition) $1 - \varepsilon$. If we let $\Omega_0(\varepsilon)$ denote the volume of vectors $J$ with overlap $R$ (or generalization error $\varepsilon$) before any data are presented, then, since the examples are independent, and since the Gibbs learning procedure returns a random element of the version space, each example will reduce this by a factor of $1 - \varepsilon$ on average. Thus, the average volume of compatible students with generalization error $\varepsilon$ after being presented $m$ training examples is

$$\Omega_m(\varepsilon) = \Omega_0(\varepsilon) \left(1 - \varepsilon\right)^m .$$

(5)

Recall that $(1 - \varepsilon)^m = \exp\left(m \ln\left(1 - \varepsilon\right)\right)$.

**Results from the traditional SM approach.** In this approach to SM, for this problem, generalization is characterized by the volume $\Omega_m(\varepsilon)$. In more detail, it is controlled by the balance between an energy and an entropy, where entropy density $s(\varepsilon)$ refers to the logarithm of the volume $s(\varepsilon) = \frac{1}{N} \ln \Omega_0(\varepsilon)$ (i.e., it is not the thermodynamic entropy, which arises in other contexts) and energy $e(\varepsilon)$ refers to the penalty due to incorrect predictions $e(\varepsilon) = \alpha \ln(1 - \epsilon)$. This is mathematically described by the extremum condition for a combination of the energy and entropy terms.[17]

- *Continuous perceptron.* For the *continuous perceptron*, we have that

$$\Omega_0(\varepsilon) \quad \sim \quad \exp\left(\frac{N}{2} \left[1 + \ln(2\pi) + \ln \sin^2(\pi\varepsilon)\right]\right)$$

$$\Omega_m(\varepsilon) \quad \sim \quad \exp\left(N \left[\frac{1}{2}\left(1 + \ln(2\pi)\right) + \frac{1}{2} \ln \sin^2(\pi\varepsilon) + \alpha \ln(1 - \varepsilon)\right]\right).$$

(6)

From this, it follows that the entropy behaves as

$$s(\varepsilon) \quad \sim \quad \frac{1}{2}\left[1 + \ln(2\pi) + \ln \sin^2(\pi\varepsilon)\right]$$

$$\sim \quad \ln(\varepsilon) \quad \text{(for small } \varepsilon \text{ or large } \alpha\text{).}$$

Observe that the entropy slowly diverges to $-\infty$, as $\varepsilon \to 0$ or as $R \to 1$. Since the examples are independent, the energy behaves as

$$e(\varepsilon) \quad \sim \quad -\alpha \ln(1 - \varepsilon)$$

$$\sim \quad \alpha\varepsilon \quad \text{(for small } \varepsilon \text{ or large } \alpha\text{).}$$

Due to the exponential in Eqn. (6), in the thermodynamic limit, this quantity is dominated by the maximum value of the expression in the square brackets of Eqn. (6). Relatedly, if a student vector is chosen at random from the version space, it will with high probability be one for which the expression in the square bracket is a maximum. To determine the maximum, we consider optimizing the difference $s(\varepsilon) - e(\varepsilon)$. See Sec. V.B of (9) for a more complete discussion; but for the small $\varepsilon$ or large $\alpha$, we have that

$$0 \sim \frac{\partial}{\partial \varepsilon} \left(\ln \varepsilon - \alpha\varepsilon\right) = \frac{1}{\varepsilon} - \alpha.$$

(7)

From this, we obtain the scaling

$$\varepsilon \sim \frac{1}{\alpha},$$

(8)

showing the smooth decrease in the generalization error with increasing number of examples. This is in accordance with PAC/VC theory, and it is illustrated in Figure 3(a).

- *Ising perceptron.* For the discrete *Ising perceptron*, we see something quite different. (Recall that in this case $J_i = \pm 1$ and $T_i = \pm 1$.) In this case, we have that

$$\Omega_0(\varepsilon) \quad \sim \quad \exp\left(N \left[-\frac{1 - R}{2} \ln\left(\frac{1 - R}{2}\right) - \frac{1 + R}{2} \ln\left(\frac{1 + R}{2}\right)\right]\right)$$

$$\Omega_m(\varepsilon) \quad \sim \quad \exp\left(N \left[-\frac{1 - R}{2} \ln\left(\frac{1 - R}{2}\right) - \frac{1 + R}{2} \ln\left(\frac{1 + R}{2}\right) + \alpha \ln\left(1 - \varepsilon\right)\right]\right), \quad (9)$$

---

[17]The precise results are quite a bit more complex than our simple summary suggests, as they involve the quenched versus annealed approximation (the latter, as in Eqn. (5)), replica-based techniques, connections with spin glasses, etc. See (9; 30; 31) for details. Replica techniques are often described as "non-rigorous," since they involve an interchange of limits. The issues arise because, when applying steepest descents approximation at one step of the method, one has to check the stability of the saddle point in the complex plane. In general, a rigorous justification is not available, but see (39) for a justification in certain cases.

where recall that $R = \cos(\pi\varepsilon)$. From this it follows that the entropy behaves as

$$s(\varepsilon) \quad \sim \quad -\frac{1 - \cos(\pi\varepsilon)}{2}\ln\left(\frac{1 - \cos(\pi\varepsilon)}{2}\right) - \frac{1 + \cos(\pi\varepsilon)}{2}\ln\left(\frac{1 + \cos(\pi\varepsilon)}{2}\right)$$

$$\sim \quad -\frac{\pi^2}{2}\varepsilon^2\ln(\varepsilon) \quad \text{(for small $\varepsilon$ or large $\alpha$).} \tag{10}$$

Observe that the entropy approaches zero as $\varepsilon \to 0$ or as $R \to 1$, meaning that there is exactly one state with $R = 1$. Since the examples are independent, the energy behaves as

$$e(\varepsilon) \quad \sim \quad -\alpha\ln(1 - \varepsilon) \tag{11}$$

$$\sim \quad \alpha\varepsilon \quad \text{(for small $\varepsilon$ or large $\alpha$).} \tag{12}$$

Following the same asymptotic analysis as with the continuous perceptron, we again consider minimizing $s(\varepsilon) - e(\varepsilon)$ by exploiting the first order condition. See Sec. V.D of (9) for a more complete discussion; but for the small $\varepsilon$ or large $\alpha$, we have that

$$0 \sim \frac{\partial}{\partial\varepsilon}\left(-\frac{\pi^2}{2}\varepsilon^2\ln(\varepsilon) - \alpha\varepsilon\right) = -\pi^2\varepsilon\ln(\varepsilon) - \alpha. \tag{13}$$

For small-to-moderate values of $\alpha$, i.e., when not too much data has been presented, this expression has a solution; but for large values of $\alpha$, this equation has no solution, indicating that the optimal value of of the expression is not inside the interval $\varepsilon \in [0, 1]$ (or $R \in [-1, 1]$), but instead at the boundary $\varepsilon = 0$ (or $R = 1$). From this, we should *not* expect a continuous and smooth decrease of the generalization error with increasing training set size, and in general there is a *discontinuous change* (drop if $\alpha$ is increasing and jump if $\alpha$ is decreasing) in $\varepsilon$ as a function of $\alpha$ at a critical value $\alpha_c$. This is not described even qualitatively by PAC/VC theory, and it is illustrated in Figure 3(c).

To summarize, the behavior of the continuous perceptron is quite simple, exhibiting the intuitive smooth decrease in generalization error with increasing data. For the discrete Ising perceptron, when the only control parameter is $\alpha$, which determines the amount of data, the generalization behavior is more complex. *In this case, there is a one-dimensional phase diagram, and depending on the value of $\alpha$, there are two phases in which the learning system can reside—one in which the generalization is large and smoothly decreasing with increasing $\alpha$, and one in which it is small or zero—and there is a discontinuous change in the generalization error between them.*

This entire discussion has focused on the simple case of realizable learning with the zero-temperature Gibbs learning rule. In general, the learning algorithm may not find a random point from the version space, the problem may not be realizable, etc., and in these cases there are additional control parameters, e.g., a temperature parameter $\tau$, e.g., to avoid reproducing the training data exactly. *In this case, the qualitative properties we have been discussion remain, but since there are two control parameters the phase diagram becomes two-dimensional, and one can have non-trivial behavior as a function of both $\alpha$ and $\tau$.* The full two-dimensional phase diagram of the discrete Ising perceptron is shown in Figure 3(d), where it shows—depending on the values of $\alpha$ and $\tau$—a phase of perfect generalization, a phase of poor generalization, a (mean field) spin glass phase, metastable regimes, etc. For completeness, the trivial two-dimensional phase diagram of the continuous perceptron (it is trivial since there is only one phase, where generalization varies continuously with $\alpha$ and $\tau$ in intuitive ways) is shown in Figure 3(b). See (9) for details.

**Results from the rigorous SM approach.** In this approach to the SM theory of learning, generalization is also characterized by a competition between an entropy-like term and an energy-like term.[18][19] In addition to providing an alternate for those with a preference for rigorous results, this approach provides intuitive pictorial explanations of the results we have observed in Figures 1(a), 1(b), 2, 3(c), and 3(d).

---

[18]Note that this resembles minimizing a Helmhotlz Free Energy $F = E - TS$, with $T = 1$.

[19]The precise results are quite a bit more complex than our simple summary suggests. See (11) for details. In particular, this approach essentially uses an ensemble (microcanonical, as opposed to the more common canonical) that permits the use of annealed theory to obtain rigorous upper bounds, rather than just approximations, and to do so for all empirical error minimization algorithms, including but not limited to zero-temperature Gibbs learning.

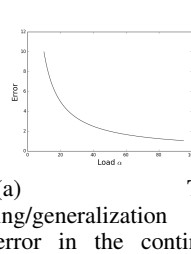 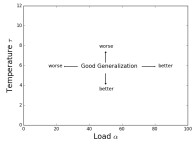 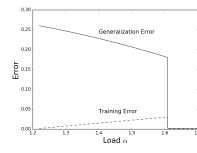 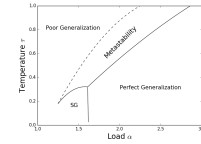

(a) Train-
ing/generalization
error in the continuous
perceptron.
(b) Learning phases in
$\tau$-$\alpha$ plane for the contin-
uous perceptron.
(c) Train-
ing/generalization
error in the discrete Ising
perceptron.
(d) Learning phases in
$\tau$-$\alpha$ plane for the discrete
Ising perceptron.

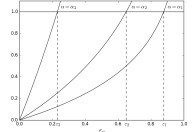 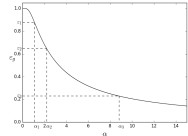 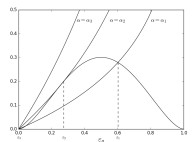 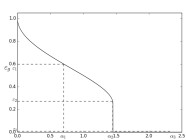

(e) Rightmost intersection
point for $s(\epsilon) = 1$ for
three values of $\alpha$.
(f) Learning curve
corresponding to entropy-
energy competition for
$s(\epsilon) = 1$.
(g) Rightmost intersec-
tion point for non-trivial $s(\epsilon)$ for three values of $\alpha$.
(h) Learning curve
corresponding to entropy-
energy competition for
non-trivial $s(\epsilon)$.

Figure 3: Training/generalization error and learning phases for the continuous and Ising perceptron; and entropy-energy trade-offs for different entropy functions.

Recall that the version space $V(\mathcal{S}) \subseteq \mathcal{F}$ is the set of all functions that are consistent with the target function $T$ on the sample $\mathcal{S}$, i.e., it is a sample-dependent subclass of $\mathcal{F}$. Also of interest is the $\epsilon$-ball around the target function, defined to be the set of functions with generalization error $\varepsilon$ not larger than $\epsilon$, i.e., $B(\epsilon) = \{h \in \mathcal{F} : \varepsilon(h) \le \epsilon\}$, which is a sample-independent subclass of $f$. Both $V(\mathcal{S})$ and $B(\epsilon)$ contain the target function $f$; and $V(\mathcal{S}) \subseteq B(\epsilon)$ implies that any consistent $h$ has generalization error $\varepsilon(h) \le \epsilon$. Thus, lower bounds on $\delta = \mathbf{Pr}\left[V(\mathcal{S}) \subseteq B(\epsilon)\right]$, where the probability is taken over the $m$ independent draws from $D$ used to obtain $\mathcal{S}$, or equivalently upper bounds on $1 - \delta = \mathbf{Pr}\left[V(\mathcal{S}) \not\subseteq B(\epsilon)\right]$, provide bounds on the generalization error $\varepsilon = \varepsilon(h)$ of *any* consistent learning algorithm outputting $h \in V(\mathcal{S})$.

From Eqn. (5), the probability that a function $h$ with generalization error $\varepsilon(h)$ remains in $V(\mathcal{S})$ after $m$ example is $\mathbf{Pr}\left[h \in V(\mathcal{S})\right] = (1 - \varepsilon(h))^m$. If we let $\overline{B(\epsilon)} = \mathcal{F} - B(\epsilon)$ denote the functions in $\mathcal{F}$ with generalization error greater than $\epsilon$, then

$$\mathbf{Pr}\left[V(\mathcal{S}) \not\subseteq B(\epsilon)\right] = \mathbf{Pr}\left[\exists h \in \overline{B(\epsilon)} : h \in V(\mathcal{S})\right] \le \sum_{h \in \overline{B(\epsilon)}} \mathbf{Pr}\left[h \in V(\mathcal{S})\right] = \sum_{h \in \overline{B(\epsilon)}} (1 - \varepsilon(h))^m.$$

If the failure probability $\delta$ rather than the error value $\epsilon$ is fixed, then it follows that if $h \in \mathcal{F}$ is any function consistent with the $m$ random examples of a target function in $\mathcal{F}$, then with probability at least $1 - \delta$, we have that

$$\varepsilon(h) = \min\left\{\epsilon : \sum_{h \in \overline{B(\epsilon)}} (1 - \varepsilon(h))^m \le \delta\right\}. \tag{14}$$

That is, the generalization error $\varepsilon(h)$ is given by a sum of quantities over $\overline{B(\epsilon)}$, and one wants to minimize $\epsilon$ in this expression to obtain improved bounds.

As a straw-man bound, assume, for simplicity, that $|\mathcal{F}| < \infty$. Then,

$$\sum_{h \in \overline{B(\epsilon)}} (1 - \varepsilon(h))^m \le \sum_{h \in \overline{B(\epsilon)}} (1 - \epsilon)^m \le |\mathcal{F}| (1 - \epsilon)^m, \tag{15}$$

from which it follows that any consistent $h$ satisfies $\varepsilon(h) \le \frac{1}{m} \ln(|\mathcal{F}|/\delta)$, with probability at least $1 - \delta$. This PAC/VC-like bound does not depend on the distribution $D$ or the target function $T$, and

it depends on $\mathcal{F}$ only via $|\mathcal{F}|$. It can, however, be very weak. (The PAC bound holds for all $h$, but it does not guarentee that the algorithm finds the best $h$.) In particular, we can have values of $\varepsilon(h)$ that are much larger than $\epsilon$; and, if we let $Q_j = |\{f' \in \mathcal{F} : \varepsilon(f') = \epsilon_j\}|$ be the number of functions in $\mathcal{F}$ with generalization error exactly $\epsilon_j$, then in general $Q_j \ll |\mathcal{F}|$.

More refined upper bounds on the left hand side of Eqn. (15) can be obtained by keeping track of errors $\epsilon_j$ (an energy) and the number of hypotheses achieving that error $Q_j$ (an entropy). Let $r \leq |\mathcal{F}|$ be the number of values that the error can assume. Since $\sum_{j=1}^r Q_j = |\mathcal{F}|$ and since $\sum_{h \in \overline{B(\epsilon_i)}} (1 - \varepsilon(h))^m = \sum_{j=i}^m Q_j(1 - \epsilon_j)^m$, one can show that Eqn. (14) becomes

$$\varepsilon(h) = \min\left\{\epsilon_i : \sum_{j=i}^r Q_j \left(1 - \epsilon_j\right)^m \leq \delta\right\}. \tag{16}$$

If, instead of considering a fixed $\mathcal{F}$, we consider a parametric class of functions, $\mathcal{F}_1, \mathcal{F}_2, \ldots, \mathcal{F}_N, \ldots$, which is needed to obtain non-trivial results, then we can rewrite the expression

$$\sum_{j=i}^{r^N} Q_j^N \left(1 - \epsilon_j^N\right)^m = \sum_{j=i}^{r^N} \exp\left(\log Q_j^N + m \left(1 - \epsilon_j^N\right)\right) \tag{17}$$

$$\leq \sum_{j=i}^{r^N} \exp\left(N \left[s(\epsilon_j^N) + \alpha \log(1 - \epsilon_j^N)\right]\right). \tag{18}$$

In Eqn. (17), for each term in the sum, $\log Q_j^N$ is positive, and $m \log(1 - \epsilon_j^N)$ is negative. If $m$ is such that $\log Q_j^N \gg -m \log\left(1 - \epsilon_j^N\right)$, for all $j$, then the value of $i$ in Eqn. (16) must be increased and we cannot give a non-trivial error bound. If, on the other hand, $m$ is such that $\log Q_j^N \ll -m \log\left(1 - \epsilon_j^N\right)$, for all $j$, then the error should be close to 0 (which, by Eqn. (16), implies small generalization error). More interesting is the intermediate regime where there is a non-trivial trade-off. To express this trade-off as a univariate optimization problem, replace $-m \log(1 - \epsilon_j^N)$ with the *energy (density) function* $e(\epsilon) = -m \log(1 - \epsilon)$, and define an *entropy (density) bound function* $s(\epsilon)$ as $\frac{1}{N} \log Q_j^N \leq s(\epsilon_j^N)$, for all $j \in \{1, \ldots, r\}$, in the appropriate limit. This leads to Eqn. (18), where $\alpha = m/N$.

To obtain generalization bounds, let $m, N \to \infty$ in such a way that $\alpha = \frac{m}{N}$ is fixed. Then, define $\epsilon^* \in [0, 1]$ to be the largest value of $\epsilon \in [0, 1]$ such that $s(\epsilon) \geq -\alpha \log(1 - \epsilon)$, defined to be 1 if $s(\epsilon) > -\alpha \log(1 - \epsilon)$, for all $\epsilon \in [0, 1]$. The main generalization bound of (11) states roughly that if we only sum terms in Eqn. (18) for which $\epsilon > \epsilon^* + \epsilon^\tau$, where $\epsilon^\tau > 0$ is arbitrary, then in this thermodynamic limit the sum equals 0, and thus in this limit we can bound generalization error by $\epsilon^* + \epsilon^\tau$. More precisely,

$$\mathbf{Pr}\left[V(\mathcal{S}) \subseteq B(\epsilon^* + \epsilon^\tau)\right] \to 1$$

To understand this in terms of the trade-off between entropy and energy, observe that $s(\epsilon)$ and $-\alpha \log(1 - \epsilon)$ are non-negative functions, and that $0 = -\alpha \log(1 - \epsilon) \leq s(\epsilon)$ for $\epsilon = 0$. Thus, $\epsilon^*$ is the right-most crossing point of these two functions. That is, it is the error value above which the energy term always dominates the entropy term. The idea then is that if $s(\epsilon) < -\alpha \log(1 - \epsilon)$, then $\exp\left(N \left[s(\epsilon) + \alpha \log(1 - \epsilon)\right]\right) \to 0$ in the thermodynamic limit.

Applied to the continuous perceptron and the Ising perceptron, we obtain the following.

- *Continuous perceptron.* For the *continuous perceptron*, an entropy upper bound of $s(\epsilon) = 1$ can be used. This is shown in Figure 3(e), along with the plots of $-\alpha \log(1 - \epsilon)$ for three different values of $\alpha$. In this case, if the rightmost intersection points are plotted as a function of $\alpha$, then one obtains Figure 3(f), which plots the learning curve corresponding to the energy-entropy competition of Figure 3(e). This figure shows a gradual smooth decrease of $\varepsilon$ with increasing $\alpha$, consistent with the results from Eqn. (8), and in accordance with PAC/VC theory.

- *Ising perceptron.* For the *Ising perceptron*, it can be shown that an entropy upper bound is $s(\epsilon) = \mathcal{H}(\sin^2(\pi\epsilon/2))$, where $\mathcal{H}(x) = -x \log x - (1 - x) \log(1 - x)$. The function $s(\epsilon)$ is shown in Figure 3(g), along with the plots of $-\alpha \log(1 - \epsilon)$ for three different values of $\alpha$. Observe that this form of $s(\epsilon)$ is consistent with Eqn. (10), for small values of $\epsilon$. That is, there are very few

configurations that have energy slightly greater than the minimum value of the energy, and thus the entropy density $s(\epsilon)$ is very small for these values of the energy $\epsilon$. In this case, if the rightmost intersection points are plotted as a function of $\alpha$, then one obtains Figure 3(h), which plots the learning curve corresponding to the energy-entropy competition of Figure 3(g). Importantly, for smaller values of $\alpha$, i.e., when working with only a modest amount of data, the rightmost crossover point is obtained at a non-zero value, and it decreases gradually as $\alpha$ is increased. Then, as more data are obtained, one hits a critical value of $\alpha$, and the plot of $s(\epsilon)$ and $-\alpha \log(1 - \epsilon)$ do *not* intersect, except at the boundary 0. At this critical value of $\alpha$, the plot in Figure 3(h) decreases suddenly to 0; and for larger values of $\alpha$, the minimum is given at the boundary. This non-smooth decrease of $\varepsilon$ with $\alpha$ is not described even qualitatively by PAC/VC theory, but it is consistent with the results from Eqn. (13), which show that the expression for the first order condition has a solution only for small-to-moderate values of $\alpha$.

## A.4 Evidence for this in more complex networks

What, one might ask, is the reason to believe that these very idealized models are appropriate to understand large, realistic DNNs?

To answer this question, recall that there is a large body of theoretical and empirical work that has focused on the so-called loss (or penalty, or energy) surfaces of NNs/DNNs (80; 81; 82; 17; 19). Figure 3 of (17) is particularly interesting in this regard. In that figure, the authors present a histogram count or entropy as a function of the loss or energy of the model, and they argue that since spin glasses exhibit similar properties, there is a connection between NNs/DNNs and spin glasses. In fact, the results presented in that figure are consistent with the much weaker hypothesis (than a spin glass) of the random energy model (REM) (83). The REM is the infinite limit of the $p$-spin spherical spin glass, which is the model analyzed by (17). It is known that the REM exhibits a transition in its entropy density at a non-zero value of the temperature parameter $\tau$, at which point the entropy vanishes; see, e.g., Chapter 5 of (75). That is, above a critical value $\tau_c$, there is a relatively large number of configurations, and below that critical value $\tau_c$, there is a single configuration (or constant number of configurations). As described for the Ising perceptron, and as opposed to the continuous perceptron, this phenomenon of having a small entropy $s(\varepsilon)$ for configurations with loss $\varepsilon$ slightly above the minimum value is the mechanism responsible for the complex learning behavior we have been discussing. This was illustrated analytically in Eqn. (13), in contrast with Eqn. (7); and it was illustrated pictorially in Figure 3(g), in contrast with Figure 3(e). This and related evidence suggests the obvious conjecture that "every" DNN exhibits this sort of phenomenon.

## A.5 Mechanisms to implement regularization

What, one might ask, is the connection between this discussion, and in particular the observation in Figure 1(c) that early stopping is a mechanism to implement regularization in the VSDL model, and other popular ways to implement regularization?

To answer this question, recall the Tikhonov-Phillips method (84), as well as the related TSVD method (85), for solving ill-posed LS problems. Given a matrix $A \in \mathbb{R}^{n \times p}$ and a vector $b \in \mathbb{R}^n$, one wants to find a vector $x \in \mathbb{R}^p$ such that $Ax = b$. A naïve solution involves computing $\hat{x} = A^{-1}b$, but there are a number of subtleties that arise. First, if $n > p$, then in general there will not exist such a vector $x$. One alternative is to consider the related LS problem

$$\hat{x} = \text{argmin}_x \|Ax - b\|_2^2, \tag{19}$$

the solution to which is $\hat{x} = \left(A^T A\right)^{-1} A^T b$. Second, if $A$ is rank-deficient, then $\left(A^T A\right)^{-1}$ may not exist; and even if it exists, if $A$ is poorly-conditioned, then the solution $\hat{x}$ computed in this way may be extremely sensitive to $A$ and $b$. That is, it will overfit the (training) data and generalize poorly to new (test) data. The Tikhonov-Phillips solution was to replace Problem (19) with the related problem

$$\hat{x} = \text{argmin}_x \|Ax - b\|_2^2 + \lambda \|x\|_2^2, \tag{20}$$

for some $\lambda \in \mathbb{R}^+$, the solution to which is

$$\hat{x} = \left(A^T A + \lambda^2 I\right)^{-1} A^T b. \tag{21}$$

The TSVD method replaces Problem (19) with the related problem

$$\hat{x} = \operatorname{argmin}_x \|A_k x - b\|_2^2, \tag{22}$$

where $A_k \in \mathbb{R}^{n \times p}$ is the matrix obtained from $A$ by replacing the bottom $p - k$ singular values with 0, i.e., which is the best rank-$k$ approximation to $A$. The solution to Problem (22) is given by

$$\hat{x} = A_k^+ b, \tag{23}$$

where $A_k^+$ is the Moore-Penrose generalized inverse of $A$.

This should be familiar, but several things should be emphasized about this general approach.

• First, the value of the control parameter $\lambda$ controls the radius of convergence of the inverse of $A^T A + \lambda^2 I$ (i.e., of the linear operator used to compute the estimator $\hat{x}$ in the Tikhonov-Phillips approach). Similarly, the value of the control parameter $k$ restricts the domain and range of $A_k$ (i.e., of the linear operator used to compute the estimator $\hat{x}$ in the TSVD approach).

• Second, *one can* always *choose a value of $\lambda$ (or $k$) to prevent overfitting, potentially at the expense of underfitting. That is, one can* always *increase the control parameter $\lambda$ (or decrease the control parameter $k$) enough to prevent a large difference between training and test error, even if this means fitting the training data very poorly. This is due to the linear structure of $A^T A + \lambda^2 I$ (and of $A_k$). For non-linear dynamical systems, or for more arbitrary linear dynamical systems, e.g., NNs from the 80s/90s or our VSDL model or realistic DNNs today, there is simply no reason to expect this to be true.*

• Third, both approaches generalize to a wide range of other problems, e.g., by considering objectives of the form $\hat{x} = \operatorname{argmin}_x f(x) + \lambda g(x)$ that generalize the bi-criteria of Problem (19), or by considering objectives such as SVMs that generalize the domain/range-restricted nature of Problem (22). In these cases, the closed-form solution of Eqns. (21) and (23) are not applicable, but it is still the case that one can always increase a control parameter ($\lambda$, the number $k$ of support vectors, etc.) enough to prevent overfitting, even if this means underfitting. Indeed, much of statistical learning theory is based on this idea (28).

• Fourth, it was well-known historically, e.g., in the 80s/90s, that these "linear" regularization approaches did *not* work well on NNs. Indeed, the main approach that did seem to work well was the early stopping of iterative algorithms that were used to train the NNs. This early stopping approach is sometimes termed implicit (rather than explicit) regularization (86; 87; 88; 7), presumably since when one tries to reduce the machine learning problem to a related optimization objective, then $\lambda$ and $k$ seem to be more natural or "fundamental" control parameters than the number of iterations (41; 42; 43). In general, when one does not do this reduction, there is no reason for this to be the case.

A complementary view of regularization arises when one considers learning algorithms that are defined operationally, i.e., as the solution to some well-defined iterative algorithm or more general dynamical system, without having a well-defined objective that is intended to be optimized exactly. In certain very special cases (e.g., for essentially linear or linearizable problems (86; 89; 87)), one can make a precise connection with the Tikhonov-Phillips/TSVD, but in general one should not expect it. From the perspective of our main results, several things are worth noting.

First, dynamics that naturally lead to the SM approach to generalization typically do not optimize linear or convex objectives, but they do have the form of a stochastic Langevin type dynamics, which in turn lead to an underlying Gibbs probability distribution (9; 30; 11; 31). Thus, far from being arbitrary dynamical systems, they have a form that is particularly well-suited to exploiting the connections with SM to obtain relatively simple generalization bounds. This dynamics has strong connections with stochastic dynamics, e.g., defined by SGD, that are used to train modern DNNs, suggesting that the SM approach to generalization can be applied more broadly.

Second, more general dynamical systems also have phases, phase transitions, and phase diagrams, where a phase is defined operationally as the set of inputs that get mapped to a given fixed point under application of the iterated dynamics, and where a phase transition is a point in parameter space where nearby points get mapped to very different fixed points (or a fixed point and and some other structure). *For general dynamical systems, however, there is not structure such as that ensured by the thermodynamic limit that can can be used to obtain generalization bounds, and there is*

*no reason to expect that control parameters of the dynamical system can serve as regularization parameters.*

Finally, we should comment on what is, in our experience, an intuition held by many researchers/practitioners, when adding noise to a system, e.g., by randomizing labels or shuffling pixel values. That is, the idea/hope: that there exists some value of some regularization parameter that will always prevent overfitting, even if that means severely underfitting; and that the quality of the generalization, i.e., the difference between training and test error, will vary smoothly with changes in that regularization parameter. There are likely several reasons for this hope. First, when one can reduce the generalization problem to an optimization problem of the form $\hat{x} = \text{argmin}_x f(x) + \lambda g(x)$, then one can clearly accomplish this simply by increasing $\lambda$. Second, upper bounds provided by the popular PAC/VC approach are smooth, and in certain limits the actual quantities being bounded are also smooth. Third, it is easier to think about and reason about the limit defined by one quantity diverging than the limit (if it exists) of two quantities diverging in some well-defined manner. *Our results in Section 3 illustrate that—and indeed a major conclusion from the SM approach to generalization is that—this intuition is in many cases simply incorrect.* It is common in ML and mathematical statistics to derive results for linear systems and then extend them to nonlinear systems by assuming that the number of data points is very large and/or that various regularity conditions hold. Our results here and other empirical results for NNs and DNNs suggest that such regularity conditions often do not hold. The consequences of this realization remain to be explored.

