# OpenReview forum: "Rethinking generalization requires revisiting old ideas: statistical mechanics approaches and complex learning behavior"
_ICLR.cc/2018/Conference — Reject_

### Official Review · AnonReviewer3 · 2017-11-16
**Fascinating but unconvincing.**

**Rating:** 3
**Confidence:** 3

**Review:**

The authors suggest that ideas from statistical mechanics will help to understand the "peculiar and counterintuitive generalization properties of deep neural networks." The paper's key claim (from the abstract) is that their approach "provides a strong qualitative description of recently-observed empirical results regarding the inability of deep neural networks not to overfit training data, discontinuous learning and sharp transitions in the generalization properties of learning algorithms, etc." This claim is restated on p. 2, third full paragraph.

I am sympathetic to the idea that ideas from statistical mechanics are relevant to modern learning theory. However, I do not find this paper at all convincing. I find the paper incoherent: I am unable to understand the argument for the central claims. On the one hand, the paper seems to be written as a "response" to Zhang et al.'s "Understanding Deep Learning Requires Rethinking Generalization", (henceforth Z): the introduction mentions Z multiple times, and the title of this work refers to Z. On the other hand, none of the issues raised by Z are (as far as I can tell) addressed in any substantial way by this paper. In somewhat more detail, this work discusses two major observations:

1. Neural nets can easily overtrain, even to random data.
2. Popular ways to regularize may or may not help.

Z certainly observes 1 and arguably observes 2. (I'd argue against, see below, but it's at least arguable.) I do not see how this paper addresses either observation. Instead, what the statistical mechanics (SM) approach seems to do is explain (or predict) the existence of phase transitions, where we suddenly go from a regime of poor generalization to good generalization or vice versa. However, neither Z nor, as far as I can tell, any other reference given here, suggests that these phase transitions are frequently observed in modern deep learning. The most relevant bit from Z is Figure 1c, which suggests that as the noise level is increased (corresponding to alpha decreasing in this paper), the generalization error increases smoothly. This seems to be in direct contradiction to the predictions made by the theories presented here.

If the authors wish to hold to the claim that their work "can provide a qualitative explanation of recently-observed empirical properties that are not easily-understandable from within PAC/VC theory of generalization, as it is commonly-used in ML" (p. 2), it is absolutely critical that they be more specific about which specific observations from which papers they think they are explaining. As written, I simply do not see which actual observations they think they explain.

In observation 2, the authors suggest that many popular ways to implement regularization "do not substantially improve the situation". A careful reading of Z (and this was corroborated by discussion with the authors) is that Z observed that regularization with parameters commonly used in practice (or, put differently, regularization parameters that led to the highest holdout accuracy in other papers) still led to substantial overtraining on noisy data. I think it is almost certainly true (see below for more discussion) that much larger values of regularization can prevent overfitting, at the cost of underfitting. It's also worth noting that Z agrees with basically all practitioners that various regularization techniques can make an important difference to practitioners who want to minimize test error; what they don't do (at least at moderate values) is *qualitatively* destroy a network's ability to overfit to noise. It is unclear to me how this paper explains observation 2 (see below for extensive discussion).

I don't actually understand the first full paragraph on p. 2 well. It is true that we can always avoid overtraining by tuning regularization parameters to get better generalization *error* (difference beween train and test) on the test data set (but possibly worse generalization accuracy); the rest of the paper seems to take the opposite side on this. A Gaussian kernel SVM with a small enough bandwidth and small enough regularization parameter can also overfit to noise. The argument needs to be sharpened here.

I find the discussion of noise at the bottom of p. 2 confusing. The authors describe tau "having to do with noise in the learning process", but then suggest that "adding noise decreases the effective load." This is the first time noise is really talked about, and it seems like maybe noise in the data is about alpha, but noise in the "learning process" is about tau? This should be clarified.

On p. 3, the authors refer to "the two parameters used by Z and many others." I am honestly not sure what's being referred to here. I just reread Z and I don't get it.  What two parameters are used by Z?

p. 3, figure. The authors should be clear about what recent (ideally widely-discussed) experimental results look anything like this figure. I found nothing in Z et al. In Appendix A.4, there is a mention of Figure 3 of Chromanska et al. 2014; that figure also seems to be totally consistent with smooth transitions and does not (to me) present any obvious evidence of a sharp phase transition. (In any case, the main paper should not rely heavily on the appendix for its main empirical evidence.)

p. 3, figure 1a. What is essential in this figure? A single phase transition? That the error be very low on the r.h.s. of the phase transition (probably not that, judging from the related models in the
Appendix).

p. 3, figure 1b/c. What does SG stand for? As far as I can tell it's never discussed.

p. 4. "Thus, an important more general insight from our approach is that --- depending strongly on details of the model, the specific details of the learning algorithm, the detailed properties of the data and their noise etc. --- going beyond worst-case bounds can lead to a rich and complex array of manners in which generalization can depend on the control parameters of the ML process." This is well-known to all practitioners. This paper does not seem to offer any specific testable explanations or predictions of any sort. I certainly agree that the study of SM models is "interesting", but what would
make this valuable would be a more direct analogy, a direct explanation of some empirical phenomenon.

Section 2 in general. The authors discuss a couple different types of observations: (1) "strong discontinuities in generalization performance as a function of control parameters" aka phase transitions, and (2) generalization performance can depend sensitively on details of the model, details of algorithms, implicit regularization properties, detailed properties of data and noise, etc." (1) shows up in the SM literature from the 90's discussed in Appendix A. I don't think it shows up in modern practice, and I don't think it shows up in Z. (2) is absolutely relevant to modern practitioners, but I don't see what this paper has to say about it beyond "SM literature from the 90's exhibits similar phenomena." The model introduced in Section 3 abstracts all such concerns away.

Section 3. I am not super comfortable with the idea of "Claims", especially since the 3 Claims seem to be different sorts of things. I would normally think of a "Claim" as something that could be true or false, possibly with some argument for its truth.

Claim 1 introduces a model (VSDL), but I wouldn't call this a claim, since nothing is actually "claimed." The subpoints of Claim 1 are arguably claims, but they're not introduced as such. I address these
in turn:

"Adding noise decreases an effective load alpha." The paper states "N is the effective capacity of the model trained on these data", but "effective capacity" is never defined. Certainly, if we *define* alpha = m_eff / N and *define* m_eff = m - m_rand, the (sub)claim follows, but why are those definitions good?  I *think* what's going on here is hidden in the sentence "empirical results indicate that for realistic DNNs it is close to 1. Thus, the model capacity N of realistic DNNs scales with m and not m_eff.", where "it" refers to the Rademacher complexity. Well, OK, but if we agree with that, then aren't we just *assuming* the main result of Z rather than explaining it? We're basically just stating that the models can memorize the data?

I don't really understand the point the last part of the paragraph is trying to make (everything after what I quoted above).

"Early stopping increases an effective temperature tau." I find this plausible but don't understand the argument at all. To this reader, it's just "stuff from SM I don't understand." I think the typical ML reader of this paper won't necessarily be familiar with any of "the weights evolve according to a relaxation Langevin equation", "from the fluctuation-dissipation theorem", or the reference to annealing rate schedules. Consider either explaining this more or just appealing to SM and relegating this to an appendix.

After the claim, the paper mentions that the VSDL model ignores other "knobs". This is fine for a model, but I think it's totally disingenuous to then suggest that this model explains anything about other popular ways to regularize (Observation 2 in the intro, see also my comment on Section 2). In the intro, the claim is "Other regularizations sometimes help and sometimes don't and we don't understand why" (the claim is about overfitting but it's also true for improving performance in general), which is basically true. But introducing a model which completely abstracts these things away cannot possibly explain anything about the behavior.

Claim 2 is that we should consider a thermodynamic limit where model complexity grows with data (the paper says grows with the number of parameters, I assume this is a typo). I would probably call this one an "Assumption", with some arguments for the justification. I think this is one of the most interesting and important ideas in the paper, and I don't fully understand it, even after reading the appendix. I have questions. How should / could this apply to practitioners, who cannot in general hope to obtain arbitrary amounts of data? Are we assuming that any (or all) modern DNN experiments are in the asymptotic regime? Are we assuming the experiments in Z are in this regime? Is there any relevance to the fact that in an ML problem (unlike in say a spin glass, at least as far as I know) the "complexity" of the *task* is *not* increasing with the data size, so eventually one will have seen "enough" data to "saturate" the task?  I'd love to know more.

Claim 3 is more of an "Informal Theorem" that under the model of Claim 1 and the assumption of Claim 2, the phase diagrams of Figure 1 hold. The "proof" is a reference to SM papers. This should be clarified.

Yet again, I point out that I do not know any modern large-scale NN experiments that correspond to any of the pictures in Figure 1.

There's a mention of "tau = 0 or t > 0." What is the significance of tau = 0? How should an ML reader think about this?

Section 3.2 suggests that Claim 3 (the existence of the 1 and 2d phase diagrams) "explain" Observations 1 and 2 from the Appendix. I simply do not see this.

For Observation 1, that NNs can easily overtrain, the "argument" seems to boil down to "the system is in a phase where it cannot help but overtrain." This is hardly an explanation at all. How do we know what phase these experiments were in? How do we know these experiments were in the thermodynamic limit?

For Observation 2, the authors point out that in VSDL, "the only way to prevent overfitting is to decrease the number of iterations." This seems true but vacuous: the authors introduced a model where regularization doesn't correspond to any knobs, so of course to the extent that that model explains reality, the knobs don't stop overfitting. But this feels like begging the question. If we accept the VSDL model, we'd also accept that various regularizations can't improve generalization, which goes directly against basically all practice. I guess I technically have to concede that "Given the three
claims", Observation 2 follows, but Claim 1 by itself seems to be already assuming the conclusion.

Minor writing issues:

The authors mention at least four times that reproducing others' results is not easy (p. 1 observation 1, p. 4 first paragraph, p. 4 footnote 6, last sentence of the main text). While I think this statement is true, it is quite well-known, and I suggest that the authors may simply alienate readers by harping on it here.

p. 1. "may always overtrain" is unclear. I don't know what it means. Is the claim that SOTA DNNs wll always overtrain when presented with enough data? I don't think so from the rest of the paper, but I'm not sure.

I'm a little unclear what the authors mean by "generalization error" (or "generalization accuracy", which seems to only be used on p. 2). Z use "generalization error = training error - test error". Check the appendix for consistency here too.

Replace "phenomenon" with "phenomena", at least twice where appropriate.

p. 3, first paragraph. I think the reference to the Hopfield model should be relegated to a footnote. The text "two or more such parameter holds more generally" is confusing; is it two, or is it two or more? What will I understand differently if I use more than two parameters? The next paragraph, starting with "Given these two identifications, which are novel to this work," seems odd, since we've
just seen 7+ references and a claim that they have similar parameterizations, so it's unclear what's novel.

Appendix A.5. "For non-linear dynamical systems... NNs from the 80s/90s or our VSDL model or realistic DNNs today .. there is simply no reason to expect this to be true." where "this" refers to "one can always choose a value of lambda to prevent overfitting, potentially at the expense of underfitting." I don't understand, and I also think this disagrees with the first full paragraph on p. 2. Is there some thermodynamic limit argument required here? The very next bullet states that x = argmin_x f(x) + lambda g(x) can prevent overfitting with large lambda. What's different? I'm overall not clear what's being implied here. Consider a modern DNN for classification. A network with all zero weights will have some empirical loss L(0). If I minimize, for the weights of a network w, L(w) + lambda ||w||^2, I have that L(w) + lambda ||w||^2 <= L(0) (assuming I can solve the optimization), and assuming L is non-negative, lambda ||w||^2 <= L(0), or ||w||^2 <= L(0) / lambda. So for very large lambda, I can drive ||w||^2 arbitrarily close to zero. How is this importantly different from the linear case?  What am I missing?

p. 3. "inability not to overfit." Avoid the double negative.

Intro, last paragraph. Weird section order description, with ref to Section A coming before section 4.

Footnote 2. "but it can be quite limiting." More detail needed. Limiting how?

Footnotes 3 and 4. The text says there are "technical" and "non-technical" reasons, but 3 and 4 both seem technical to me.

Appendix A.2. "on a randomly chosen subset of X." Is it really subset? Are we picking subsets uniformly at random?

---

> ### Author Response · Authors · 2017-12-09
> **Detailed response (7 of 7)**
>
>
> 42. Reviewer:
>
> p. 3, first paragraph. I think the reference to the Hopfield model should be relegated to a footnote. The text "two or more such parameter holds more generally" is confusing; is it two, or is it two or more? What will I understand differently if I use more than two parameters? The next paragraph, starting with "Given these two identifications, which are novel to this work," seems odd, since we've just seen 7+ references and a claim that they have similar parameterizations, so it's unclear what's novel.
>
> Response:
>
> We can clarify the "two or more" issue.  Basically, in a more realistic system, there may be many temperature-like knobs, e.g., number of iterations, annealing rate, batch size, etc., all of which control the "temperature" very imperfectly.  We predict a more complex version of what our VSDL model predicts.
>
> 43. Reviewer:
>
> Appendix A.5. "For non-linear dynamical systems... NNs from the 80s/90s or our VSDL model or realistic DNNs today .. there is simply no reason to expect this to be true." where "this" refers to "one can always choose a value of lambda to prevent overfitting, potentially at the expense of underfitting." I don't understand, and I also think this disagrees with the first full paragraph on p. 2. Is there some thermodynamic limit argument required here? The very next bullet states that x = argmin_x f(x) + lambda g(x) can prevent overfitting with large lambda. What's different? I'm overall not clear what's being implied here. Consider a modern DNN for classification. A network with all zero weights will have some empirical loss L(0). If I minimize, for the weights of a network w, L(w) + lambda ||w||^2, I have that L(w) + lambda ||w||^2 <= L(0) (assuming I can solve the optimization), and assuming L is non-negative, lambda ||w||^2 <= L(0), or ||w||^2 <= L(0) / lambda. So for very large lambda, I can drive ||w||^2 arbitrarily close to zero. How is this importantly different from the linear case?  What am I missing?
>
> Response:
>
> There are several comments here.  First, we can clarify the "this" confusion.  Second, if we read the reviewer's comment correctly, this does disagree with the first full paragraph on page 2.  That is our point: for non-linear dynamical systems, one gets something very different than, e.g, an SVM.  In somewhat more detail, and as we discuss in detail in the appendix, it is not just the thermodynamic limit, but, also the discontinuities in the models, which are important.  The SVM lacks the latter.  See the discussion in Chapter 10 of the Engle and van der Broeck book, which shows how to apply something like the thermodynamic limit to the VC bounds.  Third, this is independent of the thermodynamic limit, since for non-Langevin dynamics, there may not be such a thermodynamic system.  Fourth, we can clarify that the SVM/lambda issues are "the same" while NNs are very different.  If we understand the reviewer's question, then this would correspond to designing a network to work in a very high-temperature limit.  This would not perform as well, but this would be closer to the phase where PAC/VC intuition would hold.
>
> 44. Reviewer:
>
> p. 3. "inability not to overfit." Avoid the double negative.
>
> Response:
>
> Usually when people a double negative, they are imprecise or sloppy.  In this case, this is what we mean.
>
> 45. Reviewer:
>
> Intro, last paragraph. Weird section order description, with ref to Section A coming before section 4.
>
> Response:
>
> We agree.  In the arXiv version, it is a separate section, but we put it in an appendix to respect the page limit request.
>
> 46. Reviewer:
>
> Footnote 2. "but it can be quite limiting." More detail needed. Limiting how?
>
> Response:
>
> There are many ways in which it can be quite limiting.  For example, when one tries to work with complex realistic deep NNs, this separation breaks down, and ideas from PAC/VC theory do not provide even a qualitative guide to practice.
>
> 47. Reviewer:
>
> Footnotes 3 and 4. The text says there are "technical" and "non-technical" reasons, but 3 and 4 both seem technical to me.
>
> Response:
>
> We are saying that Footnote 3 is technical, meaning that there is a lot of technical stuff to deal with to apply the methods.  We are saying that Footnote 4 is non-technical, since we feel that it is primarily a "cultural" issue: some people like "rigorous" methods that lead to upper bounds, presumably due to their training; while other people are comfortable with approximate, mean field models, that lead to qualitative and quantitative predictions, but that require some additional nontrivial mathematical and numerical analysis to establish their rigor.
>
> 48. Reviewer:
>
> Appendix A.2. "on a randomly chosen subset of X." Is it really subset? Are we picking subsets uniformly at random?
>
> Response:
>
> It could be a randomly chosen subset that is drawn from either a uniform or a non-uniform distribution.

---

> ### Author Response · Authors · 2017-12-09
> **Detailed response (6 of 7)**
>
>
> 35. Reviewer:
>
> Section 3.2 suggests that Claim 3 (the existence of the 1 and 2d phase diagrams) "explain" Observations 1 and 2 from the Appendix. I simply do not see this.
>
> Response:
>
> Thanks.  Again, we will try to clarify.  In Fig 1c, we try to illustrate that, e.g., adding noise to the labels moves parallel to the X axis and can lead from the "Perfect" phase with good generalization to the "SG" phase or the "Poor" phase with bad generalization.  While this isn't substantially improved by changing many regularization parameters, it can be fixed by changing the tau, e.g., by changing the number of iterations, which is what Z observed.
>
> 36. Reviewer:
>
> For Observation 1, that NNs can easily overtrain, the "argument" seems to boil down to "the system is in a phase where it cannot help but overtrain." This is hardly an explanation at all. How do we know what phase these experiments were in? How do we know these experiments were in the thermodynamic limit?
>
> Response:
>
> The question of how to determine what phase can be subtle, basically since the learning process slows down dramatically, but it can be done by computing various overlap parameters.  See the papers above.  As for the question about the limits, that is  claim or hypothesis that we are making.  It is plausible, but the justification is after the fact.  In more detail, and somewhat more precisely, due to finite data and finite size effects, we know that the computations were not done in the thermodynamic limit.  Our claim is simply is that this is a less inappropriate (yes, we mean a double negative) and more useful limit than the limit taken when the model complexity is fixed and the amount of data grows.
>
> 37. Reviewer:
>
> For Observation 2, the authors point out that in VSDL, "the only way to prevent overfitting is to decrease the number of iterations." This seems true but vacuous: the authors introduced a model where regularization doesn't correspond to any knobs, so of course to the extent that that model explains reality, the knobs don't stop overfitting. But this feels like begging the question. If we accept the VSDL model, we'd also accept that various regularizations can't improve generalization, which goes directly against basically all practice. I guess I technically have to concede that "Given the three claims", Observation 2 follows, but Claim 1 by itself seems to be already assuming the conclusion.
>
> Response:
>
> The claim that "the only way to prevent overfitting is to decrease the number of iterations" is not vacuous.  It is one of the main observations in Z.  Although they don't describe it as such, it is clear from their empirical results that while traditional regularization knobs don't help much, there is one knob that has a strong effect of regularization, and that is the stopping time.  This was known in the 80s.  We simply point out that revisiting these old ideas can explain what is going on in much more complex computation of interest to the ICLR community.
>
> 38. Reviewer:
>
> Minor writing issues:
>
> The authors mention at least four times that reproducing others' results is not easy (p. 1 observation 1, p. 4 first paragraph, p. 4 footnote 6, last sentence of the main text). While I think this statement is true, it is quite well-known, and I suggest that the authors may simply alienate readers by harping on it here.
>
> Response:
>
> Thanks for the suggestion.  We will try to moderate the claims.  We do, however, think this is not simply complaining.  Instead, it is closely related to the thermodynamic limiting arguments.  In this limit, little details matter a lot more, and so minor details in the problem can be extremely important.
>
> 39. Reviewer:
>
> p. 1. "may always overtrain" is unclear. I don't know what it means. Is the claim that SOTA DNNs wll always overtrain when presented with enough data? I don't think so from the rest of the paper, but I'm not sure.
>
> Response:
>
> Thanks.  We agree is is slightly imprecise.  Since we weren't able to reproduce results, we weren't able to come up with a more precise version of the statement with which we are comfortable.  It certainly won't always overtrain, e.g., if we run zero steps of an iterative method.  Whether it will "always" overtrain if we try to push the boundary and get good/state-of-the-art prediction results is unclear to us.
>
> 40. Reviewer:
>
> I'm a little unclear what the authors mean by "generalization error" (or "generalization accuracy", which seems to only be used on p. 2). Z use "generalization error = training error - test error". Check the appendix for consistency here too.
>
> Response:
>
> Thanks, the literature is inconsistent, and we tried to be consistent, but we will double check.
>
> 41. Reviewer:
>
> Replace "phenomenon" with "phenomena", at least twice where appropriate.
>
> Response:
>
> Thanks, we tried to be consistent, but we will double check.

---

> ### Author Response · Authors · 2017-12-09
> **Detailed response (5 of 7)**
>
>
> 30. Reviewer:
>
> After the claim, the paper mentions that the VSDL model ignores other "knobs". This is fine for a model, but I think it's totally disingenuous to then suggest that this model explains anything about other popular ways to regularize (Observation 2 in the intro, see also my comment on Section 2). In the intro, the claim is "Other regularizations sometimes help and sometimes don't and we don't understand why" (the claim is about overfitting but it's also true for improving performance in general), which is basically true. But introducing a model which completely abstracts these things away cannot possibly explain anything about the behavior.
>
> Response:
>
> We don't think that it is "disingenuous" to do this.  Rather than asking for the most complex model we can come up with, the point of the paper is what is the simplest model that will shed insight into the problem.  As for the comment "A model which completely abstracts these things away cannot possibly explain anything about the behavior," we really don't know what to say.  If this is the case, then noone should be talking about Rademacher complexities in the first place.  Even SVMs have nothing to do with this since the hypothesis space is chosen in a data dependent way.  The question is one of a level of abstraction, and what can be learned from that abstraction.  The key point here is that SM treats learning as an "emergent" phenomena and studies the properties of learning using simple models because experience with other problems in complexity theory indicate that we can learn alot---not everything, but a lot---about complex systems without needing to model the very specific details of the network architectures.  For example, we can use simplified McCulloch and Pitts neurons and study the emergent behavior of collections of these basic objects, etc.  At a minimum, it provides a theory of learning that is worth "revisiting," which is the entire point of our paper.
>
> 31. Reviewer:
>
> Claim 2 is that we should consider a thermodynamic limit where model complexity grows with data (the paper says grows with the number of parameters, I assume this is a typo). I would probably call this one an "Assumption", with some arguments for the justification. I think this is one of the most interesting and important ideas in the paper, and I don't fully understand it, even after reading the appendix. I have questions. How should / could this apply to practitioners, who cannot in general hope to obtain arbitrary amounts of data? Are we assuming that any (or all) modern DNN experiments are in the asymptotic regime? Are we assuming the experiments in Z are in this regime? Is there any relevance to the fact that in an ML problem (unlike in say a spin glass, at least as far as I know) the "complexity" of the *task* is *not* increasing with the data size, so eventually one will have seen "enough" data to "saturate" the task?  I'd love to know more.
>
> Response:
>
> Thanks for catching this.  Technical complexities aside, we feel that one of the most interesting things there is our observation (be it a pedagogical claim or an assumptions) that the usual limiting arguments from mathematical statistics are less appropriate that the thermodynamic limit.  As for applying it to practitioners, we can imagine many possibilities, and that is for the next paper.  As for other problems in ML, while one may peek at the data to develop features, e.g., for an SVM, one arguably has a less strong dependence on the data than in NNs where one peeks at the data many many times.  That is why PAC/VC can shed light on SVMs and many other models but not NNs.  BTW, even for Tikhonov regularization, subtle properties are observed that are not captured by VC theory, e.g.:
>
> https://github.com/ryotat/ryotat.github.io/blob/master/teaching/enshu12.pdf
>
> 32. Reviewer:
>
> Claim 3 is more of an "Informal Theorem" that under the model of Claim 1 and the assumption of Claim 2, the phase diagrams of Figure 1 hold. The "proof" is a reference to SM papers. This should be clarified.
>
> Response:
>
> Thanks, we can clarify that.
>
> 33. Reviewer:
>
> Yet again, I point out that I do not know any modern large-scale NN experiments that correspond to any of the pictures in Figure 1.
>
> Response:
>
> See our comments above.
>
> 34. Reviewer:
>
> There's a mention of "tau = 0 or t > 0." What is the significance of tau = 0? How should an ML reader think about this?
>
> Response:
>
> tau>0 is a positive temperature that interpolates between tau=infinity (where "everything is random") and tau=0 (where "everything is discrete").  It has been used, e.g., to have a "relaxed" or "soft" version of combinatorial optimization problems, e.g., as solved with temperature-annealed MCMC in simulated annealing.

---

> ### Author Response · Authors · 2017-12-09
> **Detailed response (4 of 7)**
>
>
> 27. Reviewer:
>
> Section 3. I am not super comfortable with the idea of "Claims", especially since the 3 Claims seem to be different sorts of things. I would normally think of a "Claim" as something that could be true or false, possibly with some argument for its truth.
>
> Response:
>
> We agree.  We did it for pedagogical reasons.
>
> 28. Reviewer:
>
> Claim 1 introduces a model (VSDL), but I wouldn't call this a claim, since nothing is actually "claimed." The subpoints of Claim 1 are arguably claims, but they're not introduced as such. I address these in turn:
>
> "Adding noise decreases an effective load alpha." The paper states "N is the effective capacity of the model trained on these data", but "effective capacity" is never defined. Certainly, if we *define* alpha = m_eff / N and *define* m_eff = m - m_rand, the (sub)claim follows, but why are those definitions good?  I *think* what's going on here is hidden in the sentence "empirical results indicate that for realistic DNNs it is close to 1. Thus, the model capacity N of realistic DNNs scales with m and not m_eff.", where "it" refers to the Rademacher complexity. Well, OK, but if we agree with that, then aren't we just *assuming* the main result of Z rather than explaining it? We're basically just stating that the models can memorize the data?
>
> Response:
>
> We are saying that data and/or noise added to labels/data can be viewed in terms of a load parameter.  This is not assuming the main results of Z, and it says nothing about memorization or generalization.  Then, appealing to SM results, this has implications for memorization/generalization that go beyond what PAC/VC theory can say.  In addition, the parameter regime where "memorization" occurs is actually for much smaller values of the load than we are considering.  Memorization and over-training are different phenomena in the SM theory, and memorization occurs at extremely small values of the load.  This is related to point 42 below, and why we mention the Hopfield model.  The Hopfield model is a model of memorization, and we don't think that is what is going on here.
>
>
> 29. Reviewer:
>
> I don't really understand the point the last part of the paragraph is trying to make (everything after what I quoted above).
>
> "Early stopping increases an effective temperature tau." I find this plausible but don't understand the argument at all. To this reader, it's just "stuff from SM I don't understand." I think the typical ML reader of this paper won't necessarily be familiar with any of "the weights evolve according to a relaxation Langevin equation", "from the fluctuation-dissipation theorem", or the reference to annealing rate schedules. Consider either explaining this more or just appealing to SM and relegating this to an appendix.
>
> Response:
>
> Thanks for the honesty about not understanding this.  Indeed, this was predicted in the blog we cited above:
>
> https://medium.com/intuitionmachine/revisiting-deep-learning-as-a-non-equilibrium-process-9cedb93a13a2
>
> The point of this paper, and in particular the point of our more pedagogical explanation, is to highlight and explain these "simple" (simple for people trained in this area rather a different area) ideas, rather than wrapping these simple ideas into something technically complex that has epsilon novelty.  Importantly, implementing these simple ideas is quite complex, and it is appropriate for follow up work.  But, note that there are several recent papers that have begun to use these ideas.  We are happy to move this material to a main section, if it is acceptable to the PC.  As for an even more detailed explanation, we feel that it will be more appropriate for a follow-up work than an 8 page conference paper.

---

> ### Author Response · Authors · 2017-12-09
> **Detailed response (3 of 7)**
>
>
> 21. Reviewer:
>
> On p. 3, the authors refer to "the two parameters used by Z and many others." I am honestly not sure what's being referred to here. I just reread Z and I don't get it.  What two parameters are used by Z?
>
> Response:
>
> Thank you again for highlighting that you are confused by this.  We will try to clarify.  Z did a lot, and we present a very simple model of what they did.  Basically, they fit a NN.  Then they added noise to the training data (one knob, which we model as a load parameter); then they tried traditional regularization knobs (which we do not model since they didn't help substantially); then they plotted quality as a function of the number of iterations (another knob, which has a temperature interpretation, which we model as a temperature control parameter).
>
> 22. Reviewer:
>
> p. 3, figure. The authors should be clear about what recent (ideally widely-discussed) experimental results look anything like this figure. I found nothing in Z et al. In Appendix A.4, there is a mention of Figure 3 of Chromanska et al. 2014; that figure also seems to be totally consistent with smooth transitions and does not (to me) present any obvious evidence of a sharp phase transition. (In any case, the main paper should not rely heavily on the appendix for its main empirical evidence.)
>
> Response:
>
> The lack of sharpness is a finite size effect.  In particular, any paper that mentions the words "spin glass" (including in the 5th line of the Choromanska et al paper, as well as many others in the area) has these transitions, since that is fundamental to what is a spin glass.  See also several of the other papers mentioned above.  BTW, in the arXiv version, the appendix is simply another section.  We put this important information in an appendix to respect page limitation requests.  Thanks for reading it, since it is an important part of the paper.
>
> 23. Reviewer:
>
> p. 3, figure 1a. What is essential in this figure? A single phase transition? That the error be very low on the r.h.s. of the phase transition (probably not that, judging from the related models in the Appendix).
>
> Response:
>
> This should be compared with Fig 3.  There could be one or several phase transitions.  The point is that the actual generalization error does not decrease smoothly in a nice inverse polynomial way, as the PAC/VC upper bounds do.
>
> 24. Reviewer:
>
> p. 3, figure 1b/c. What does SG stand for? As far as I can tell it's never discussed.
>
> Response:
>
> Thanks for catching this.  This is "Spin Glass" phase.  We will clarify in the final version.
>
> 25. Reviewer:
>
> p. 4. "Thus, an important more general insight from our approach is that --- depending strongly on details of the model, the specific details of the learning algorithm, the detailed properties of the data and their noise etc. --- going beyond worst-case bounds can lead to a rich and complex array of manners in which generalization can depend on the control parameters of the ML process." This is well-known to all practitioners. This paper does not seem to offer any specific testable explanations or predictions of any sort. I certainly agree that the study of SM models is "interesting", but what would make this valuable would be a more direct analogy, a direct explanation of some empirical phenomenon.
>
> Response:
>
> We agree that this is well-known to practitioners.  The point is that this is not at all predicted by PAC/VC theory.  Our paper "revisits" old ideas from the SM theory of generalization that does predict this behavior in simple models.  We would love to make strong quantitative predictions on large-scale realistic models.  That seems more suited for a follow-up paper.
>
> 26. Reviewer:
>
> Section 2 in general. The authors discuss a couple different types of observations: (1) "strong discontinuities in generalization performance as a function of control parameters" aka phase transitions, and (2) generalization performance can depend sensitively on details of the model, details of algorithms, implicit regularization properties, detailed properties of data and noise, etc." (1) shows up in the SM literature from the 90's discussed in Appendix A. I don't think it shows up in modern practice, and I don't think it shows up in Z. (2) is absolutely relevant to modern practitioners, but I don't see what this paper has to say about it beyond "SM literature from the 90's exhibits similar phenomena." The model introduced in Section 3 abstracts all such concerns away.
>
> Response:
>
> For the first point, we agree; see the comments above about so-called finite size effects.  For the second point, we tried to work with the simplest model that would "explain" the results, rather than a much more complex model that would hide the essential issues.  So, it is not the case that our model "abstracts all such concerns away," it just abstracts away almost all the things that are not essential to understand the basic point.  See comments below about that.

---

> ### Author Response · Authors · 2017-12-09
> **Detailed response (2 of 7)**
>
>
> 15. Reviewer:
>
> If the authors wish to hold to the claim that their work "can provide a qualitative explanation of recently-observed empirical properties that are not easily-understandable from within PAC/VC theory of generalization, as it is commonly-used in ML" (p. 2), it is absolutely critical that they be more specific about which specific observations from which papers they think they are explaining. As written, I simply do not see which actual observations they think they explain.
>
> Response:
>
> Our results hold more generally, but as stated we "explain" the two main observations in the Z paper.  The key observation in the Z paper is that "Even with dropout and weight decay, Inception V3 is still able to fit [a] random training set extremely well if not perfectly" , but lacks any generalization capacity.  See their discussion and their main Table.
>
> 16. Reviewer:
>
> In observation 2, the authors suggest that many popular ways to implement regularization "do not substantially improve the situation". A careful reading of Z (and this was corroborated by discussion with the authors) is that Z observed that regularization with parameters commonly used in practice (or, put differently, regularization parameters that led to the highest holdout accuracy in other papers) still led to substantial overtraining on noisy data.
>
> Response:
>
> We are not sure what the reviewer is saying.  That is what we are saying, i.e., "do not substantially improve" = "still led to substantial overtraining".
>
> 17. Reviewer:
>
> I think it is almost certainly true (see below for more discussion) that much larger values of regularization can prevent overfitting, at the cost of under-fitting.
>
> Response:
>
> We are not so confident.  This is an empirical question, beyond the scope of this paper both for idealized NNs as well as for realistic NNs.  But we discuss this issue in detail in Appendix A.5, where we note that this intuition is true for popular ML models but not necessarily true in general.
>
> 18. Reviewer:
>
> It's also worth noting that Z agrees with basically all practitioners that various regularization techniques can make an important difference to practitioners who want to minimize test error; what they don't do (at least at moderate values) is *qualitatively* destroy a network's ability to overfit to noise. It is unclear to me how this paper explains observation 2 (see below for extensive discussion).
>
> Response:
>
> We agree.  The qualitative destruction is a statement about the thermodynamic limit, which is a statement about a limit.  For finite N, there are finite N effects.  References 9, 30; 11, and 31, as well as many others they cite, clearly show that the limiting behavior is smudged out for finite N.
>
> 19. Reviewer:
>
> I don't actually understand the first full paragraph on p. 2 well. It is true that we can always avoid overtraining by tuning regularization parameters to get better generalization *error* (difference between train and test) on the test data set (but possibly worse generalization accuracy); the rest of the paper seems to take the opposite side on this. A Gaussian kernel SVM with a small enough bandwidth and small enough regularization parameter can also overfit to noise. The argument needs to be sharpened here.
>
> Response:
>
> Thank you for highlighting that you are confused by this.  This is the central point of the argument.  The SM theory suggests that it is false that one can always do this.  I.e., that this is true for SVMs as this paragraph point outs, but that it is false for DNNs.  More precisely, if one's "control knobs" are traditional regularization parameters, then it is false.  On the other hand, if one's control knobs include the iteration count in early stopped algorithms, which have a natural interpretation in terms of temperature as we and others have argued, then one can exert this control.
>
> 20. Reviewer:
>
> I find the discussion of noise at the bottom of p. 2 confusing. The authors describe tau "having to do with noise in the learning process", but then suggest that "adding noise decreases the effective load." This is the first time noise is really talked about, and it seems like maybe noise in the data is about alpha, but noise in the "learning process" is about tau? This should be clarified.
>
> Response:
>
> Thank you for highlighting that you are confused by this.  One type of "noise" is adding noise to the data, as Z and others do.  Another type of "noise" is variability in, e.g., early-stopped SGD algorithms.  This is a very important difference, and we will clarify in the final version.

---

> ### Author Response · Authors · 2017-12-09
> **Detailed response (1 of 7)**
>
>
> Let's provide a detailed response to the least confident reviewer's points.
>
> 11. Reviewer:
>
> The authors suggest that ideas from statistical mechanics will help to understand the "peculiar and counterintuitive generalization properties of deep neural networks." The paper's key claim (from the abstract) is that their approach "provides a strong qualitative description of recently-observed empirical results regarding the inability of deep neural networks not to overfit training data, discontinuous learning and sharp transitions in the generalization properties of learning algorithms, etc." This claim is restated on p. 2, third full paragraph.
>
> I am sympathetic to the idea that ideas from statistical mechanics are relevant to modern learning theory. However, I do not find this paper at all convincing. I find the paper incoherent: I am unable to understand the argument for the central claims.
>
> Response:
>
> A recent blog has highlighted that "There are several papers that also come from those trained in a field other than statistics, that will likely not see the light of day (or rather accepted in a conference). The incomprehensibility to the reviewer trained only in statistics is grounds for rejection."  See:
>
> https://medium.com/intuitionmachine/revisiting-deep-learning-as-a-non-equilibrium-process-9cedb93a13a2
>
> We are well aware that the SM methods are quite different from popular methods in ML/DL/AI, and that some readers will find these quite different methods initially incomprehensible/incoherent.  We are trying to make these methods accessible to readers not trained in SM, since they can be used to understand the phenomena observed by Z.
>
> 12. Reviewer:
>
> On the one hand, the paper seems to be written as a "response" to Zhang et al.'s "Understanding Deep Learning Requires Rethinking Generalization", (henceforth Z): the introduction mentions Z multiple times, and the title of this work refers to Z. On the other hand, none of the issues raised by Z are (as far as I can tell) addressed in any substantial way by this paper. In somewhat more detail, this work discusses two major observations:
>
> 1. Neural nets can easily overtrain, even to random data.
> 2. Popular ways to regularize may or may not help.
>
> Z certainly observes 1 and arguably observes 2. (I'd argue against, see below, but it's at least arguable.) I do not see how this paper addresses either observation. Instead, what the statistical mechanics (SM) approach seems to do is explain (or predict) the existence of phase transitions, where we suddenly go from a regime of poor generalization to good generalization or vice versa.
>
> Response:
>
> At a superficial level, the SM approach explains/predicts phase transitions.  More generally, it describes generalization behavior, one component of which is sharp transitions, as well as which control parameters of the learning process can be used to control generalization quality.
>
> 13. Reviewer:
>
> However, neither Z nor, as far as I can tell, any other reference given here, suggests that these phase transitions are frequently observed in modern deep learning.
>
> Response:
>
> We do not claim that they are frequently observed.  Clearly, they are not.  The question Z specifically asks is whether there "is a different form of capacity control that bounds generalization error for large neural nets." Large is the key word here. Phase transitions are a limiting phenomenon, and so in any system with only a finite amount of data, there will be finite-size effects.  Plus, in practical systems, one often engineers the system to get close to but to avoid this transition, e.g., by engineering the data or the learning process to smooth out the transition.  However, our analysis suggests that phase transitions are "under the hood" and, being a limiting effect, are predicted to be more relevant in larger systems.  So it would be useful for the community to understand them better.
>
> 14. Reviewer:
>
> The most relevant bit from Z is Figure 1c, which suggests that as the noise level is increased (corresponding to alpha decreasing in this paper), the generalization error increases smoothly. This seems to be in direct contradiction to the predictions made by the theories presented here.
>
> Response:
>
> We tried but were not able to reproduce the results of the Z paper.  Our working hypothesis which we are working on testing is that this is a finite size effect.

---

> ### Author Response · Authors · 2018-01-07
> **preilimary results are available**
>
> We do have very preliminary results that indicate the presence of a phase transition in this system.
>
> We have been able to reproduce the results of the 3-layer MLP.  We identify the  phase transition by measuring the
> generalized Von Neumann matrix entropy* of layer weight matrices S(W) .   *(See PNAS August 29, 2000 u vol. 97 u no. 18 u 10101–10106)
>
>
> We can measure S(W1), S(W2), and S(W3) for each weight matrix in the network. [We did not measure S(W1) but we assume the results would be similar]
>
> We find that for a normal data set, S(W2) and S(W3) decreases slowly  (within 1-2 %) or not all with each epoch.
>
> For a data set with fully randomized labels, however, S(W2) and S(W3) both display a first order phase transition after about 20-25 epochs, changes in value by 10% or more.
>
> These very early results indicate both a phase transition, as predicted by theory, and a drop in entropy, also predicted. We have not presented them because they are both very early and we prefer to present them in a second paper, which is less pedagogical and more about numerical results.

---

### Official Review · AnonReviewer2 · 2017-11-25
**Interesting set of ideas and direction, but lack of quantitative analysis supporting the results.**

**Rating:** 6
**Confidence:** 5

**Review:**


This papers provides an interesting set of ideas related to theoretical understanding generalization properties of multilayer neural networks. It puts forward a qualitative analogy between some recently observed behaviours in deep learning and results stemming from previous quantitative statistical physics analysis of single and two-layer neural networks. The paper serves as a nice highlight into the not-so recent progress made in statistical physics for understanding of various models of neural networks. I agree with the authors that this line of work, that is not very well known in the current machine learning community, includes a number of ideas that should be able to shed light on some of the currently open theoretical questions. As such the paper would be a nice contribution to ICLR.

On the negative side, the paper is only qualitative. The Very Simple Deep Learning model that it introduces is not even a model in the physics or statistics sense, since it cannot be fit on data, it does not specify any macroscopic details. I only saw something like that to be called a *model* in experimental biology papers ... The models that are reviewed in the appendix, i.e. the continuous and Ising perceptron and the committee machine are more relevant. However, the present paper only reviews existing results about them. And even in that there are flaws, because it is not always clear from what previous works are the results taken nor is it clear how exactly they were obtained (e.g.  Fig. 2 (a) is for Ising or continuous weights? How was it computed? Why in Fig. 3(a) the training and generalization error is the same while in Fig. 3(c) they are different? What exact formulas were evaluated to obtain these figures?).

Concerning the lack of mathematical rigour in the statistical physics literature on which the authors comment, they might want to relate to a very recent work https://arxiv.org/pdf/1708.03395.pdf work that sets all the past statistical physics results on optimal generalization in single layer neural networks on fully rigorous basis by proving that the corresponding formulas stemming from the replica method are indeed correct.

---

### Official Review · AnonReviewer1 · 2017-11-26
**Interesting remarks, nice review, but maybe lack new results?**

**Rating:** 7
**Confidence:** 4

**Review:**

I find myself having a very hard time making a review of this paper,  because I mostly agree with the intro and discussion, and certainly agree that the "typical" versus "worse case" analysis is certainly an important point.  The authors are making a strong case for the use of these models to understand overfitting and generalization in deep leaning.

The problem is however that, except from advocating the use of these "spin glass" models studied back in the days by Seung, Sompolinksy, Opper and others, there are little new results presented in the paper. The arguments using the Very Simple Deep Learning (VSDL) are essentially a review of old known results --which I agree should maybe be revisited-- and the motivation to their application to deep learning stems from the reasoning  that, since this is the behavior observed in all these model, well then deep learning should behave just the same as well. This might very well be, but this is precisly the point: is it ?

After reading the paper,  I agree with many points and enjoyed reading the discussion. I found interesting ideas discussed and many papers reviewed, and ended up discovering interesting papers on arxiv as a concequence.

This is all nice, interesting, and well written, but at the end of the day, the paper is not doing too much beyond being a nice review of all ideas. While this has indeed some values, and might trigger a renewal of interested for these approaches, I will let the comity decide if this is the material they want in ICLR.

A minor comment: The generalization result of [9,11] obtained with heuristic tools (the replica method of statistical mechanics) and plotted in Fig.1 (a) has been proven recently with rigorous mathematical methods in arxiv:1708.03395

Another remark:  if deep learning is indeed well described by these models, then again so are many other simpler problems, such as compressed sensing, matrix and tensor factorization, error corrections, etc etc... with similar phase diagram as in fig. 1.  For instance gaussian mixtures are discussed in http://iopscience.iop.org/article/10.1088/0305-4470/27/6/016/and  SVM (which the authors argue should behave quite differently) methods have been treated by statistical mechanics tools in https://arxiv.org/pdf/cond-mat/9811421.pdf with similar phase diagrams. I am a bit confused what would be so special about deep learning then?

---

### Author Response · Authors · 2017-12-08
**Two reviewers were confident and positive; the third reviewer had deep misunderstandings which we address.**

Two reviewers were confident and positive. The least confident reviewer was least positive. The detailed questions (longer than our permitted response) indicate a well-intentioned reviewer, but one who has deep misunderstandings about our paper and the prior results our paper explains.

For the confident positive reviewers.

1. It is correct that we did not present "new" technically incremental results. This approach makes it easier for readers to understand ideas which may be quite unfamiliar.

2. As for SM applied to SVMs, both SM and PAC/VC and extensions can be applied to anything. The question is whether they say anything nontrivial. For SVMs, SM predicts phase transitions, and PAC/VC predicts generalization can be controlled with regularization parameters, number of support vectors, etc. For NNs, the latter is not true. This is the point of Zhang et al. (We will also use Z to refer to this.) SM can explain what is going on in Z. That is the point of our paper.

3. We do more than draw a qualitative analogy. We propose this old theory applies to new deep NNs; and we explain how/why this happens in terms of entropy (in an appendix, due to page limitations). Fig 3a and 3c highlight the key difference, expanded in Figs 3e-3h, is whether a model has nontrivial entropy properties near the minimum energy. This is common to all other models, e.g., those in Fig 2. This connection is buried in previous work. Highlighting it is an important contribution.

4. Thanks for the Barbier reference.

Next, the comments of the least confident reviewer highlight deep misunderstandings about our paper, as well as the Z paper. We expected this would happen for some readers because the work relies on established results from the statistical mechanics of learning, which may be unfamiliar to some reviewers.

These misunderstandings are going to be shared by other readers, so we are glad to have the opportunity to respond.  Some general points.

5. Our paper is about theory applied to practice. It address the Z paper, which claims that VC theory does not work at all in practice. Z shows that "Even with dropout and weight decay, Inception V3 is still able to fit [a] random training set extremely well if not perfectly", but lacks any generalization capacity. VC theory is a theory about capacity control. But it does not exhibit the behavior described in Z. SM does describe this behavior. BTW, VC theory was never expected to apply to NNs, shallow or deep. This was pointed out by Vapnik, Levin, and LeCun in 1994 ("Measuring the VC-dimension of a learning machine"):

"The extension of this work to multilayer networks faces [many] difficulties ... the existing learning algorithms can not be viewed as minimizing the empirical risk over the entire set of functions implementable by the network ... [because] it is likely ... the search will be confined to a subset of [these] functions ... The capacity of this set can be much lower than the capacity of the whole set ... [and] may change with the number of observations. This may require a theory that considers the notion of a non-constant capacity with an ‘active’ subset of functions"

This observation is true whether we are considering PAC/VC, or variants, e.g., Covering numbers, Rademacher complexity, etc. The authors of Z clearly know this. These approaches make gross assumptions that are clearly at odds with basic experimental observations (and SM theory). The value of Z is to highlight these old ideas that have largely been forgotten. The value of our paper is to remind the community about old ideas that have also largely been forgotten but that do describe this.

6. Rarely would we just add more data (m) to a deep NN. We always increase the NN size (N) too. The reason is we can capture more detailed features/information from the data.  That is, we do in practice what we argue for in the paper---take the limit of large size, with the ratio m/N fixed (as opposed to fixing m and let N increase, or vice versa, which is at odds with practice).

7. As for whether phase behavior is directly observed in deep nets: of course not. Phases arises in the thermodynamic limit. Any real system will show finite size effects, i.e., the sharp behavior will be smoothed out. This is well know and well studied.

8. As for reproducibility, Z did not provide code. There is non-current pyTorch code on github. This will take additional work.

9. On the SVM, the point is that SVMs can always be regularized to avoid over training.  For NNs, popular regularization methods can sometimes fail to prevent overtraining.  The reason is the NN is beyond the critical value of alpha where this is possible.

10. Experiments are important follow-up work. The thing to measure is the replica overlap or another order parameter of the layers in each phase. Similar work has begun by Ganguli et al:

https://arxiv.org/abs/1611.01232
https://arxiv.org/abs/1711.04735

---

### Decision · Program_Chairs · 2018-01-29
**ICLR 2018 Conference Acceptance Decision**

**Decision:**

Reject

**Comment:**

The concerns raised by AnonReviewer3 point out that, despite the effort of the authors to bridge the SM / ML divide, there is still some work to be done. The gulf between thermodynamic limits and finite effects is oft-cited in the author response. This seems to be a catch all. This gap needs to be addressed early. The authors might even suggest some open (empirical) questions looking for these phase transitions in finite systems in cases where they think engineering has placed us "not too close".